# Reliable Decision-Making via Calibration-Oriented Retrieval-Augmented Generation

**Chaeyun Jang**
KAIST
jcy9911@kaist.ac.kr

**Deukhwan Cho**
KAIST
macro_boomin@kaist.ac.kr

**Seanie Lee**
KAIST
lsnfamily02@kaist.ac.kr

**Hyungi Lee**[†]
Kookmin University
lhk2708@kookmin.ac.kr

**Juho Lee**[†]
KAIST
juholee@kaist.ac.kr

## Abstract

Recently, Large Language Models (LLMs) have been increasingly used to support various decision-making tasks, assisting humans in making informed decisions. However, when LLMs confidently provide incorrect information, it can lead humans to make suboptimal decisions. To prevent LLMs from generating incorrect information on topics they are unsure of and to improve the accuracy of generated content, prior works have proposed Retrieval Augmented Generation (RAG), where external documents are referenced to generate responses. However, previous RAG methods focus only on retrieving documents most relevant to the input query, without specifically aiming to ensure that the human user's decisions are well-calibrated. To address this limitation, we propose a novel retrieval method called Calibrated Retrieval-Augmented Generation (CalibRAG), which ensures that decisions informed by RAG are well-calibrated. Then we empirically validate that CalibRAG improves calibration performance as well as accuracy, compared to other baselines across various datasets.

## 1 Introduction

Large language models [LLMs; 1, 2, 3, 4] have demonstrated remarkable performance on numerous downstream natural language processing (NLP) tasks, leading to their widespread integration into various decision-making processes [5, 6, 7]. However, even with significant increases in model size and the expansion of training datasets, it remains infeasible for LLMs to encode all possible knowledge within their parameters. As a result, the outputs produced by LLMs may not consistently be reliable for important human decision-making processes, potentially overlooking key or hidden details. Additionally, LLMs frequently provide inaccurate or misleading information with a high degree of confidence, a phenomenon referred to as *hallucination* [8, 9], which can lead humans to make flawed decisions. In addition, Zhou et al. [7] has empirically demonstrated that human users often over-rely on LLM outputs during decision-making processes, and this over-reliance tends to increase in proportion to the model's confidence. Here, the model's confidence refers to the expression of how certain the model is when asked how confident it is in its answer. Specifically,

---

[†]Co-corresponding authors

39th Conference on Neural Information Processing Systems (NeurIPS 2025).

they have found that for answers with high confidence, users show strong over-reliance regardless of whether the answer is correct or not. These findings highlight that utilizing LLMs without proper calibration of their responses and addressing the frequent occurrence of hallucinations can lead to incorrect decisions in high-stakes tasks such as medical diagnosis and legal reasoning, potentially resulting in severe consequences [10, 11, 12].

Retrieval Augmented Generation (RAG) [13, 14, 15] has emerged as a promising method to address hallucinations, which is one of the two key issues when using LLMs in decision-making [16, 17]. Instead of generating answers directly, RAG retrieves relevant documents from external databases and uses them as an additional context for response generation. This approach supplements the information that LLMs lack, resulting in more accurate and reliable responses. However, the database cannot encompass all information, and the knowledge from world is continuously being updated. In such cases, the retriever may retrieve irrelevant documents, which can distract the LLM and lead to the generation of incorrect answers to the question [18]. Moreover, as described in Sec. 2.2, due to the LLM's overconfidence in the retrieved document, they still tend to assign high confidence to its responses even when they are incorrect.

Research on *uncertainty calibration* has aimed to address the issue of overconfident outputs in deep neural networks [19, 20, 21]. In image classification, techniques like temperature scaling have proven effective in improving calibration by adjusting logits [22, 23, 24]. However, calibrating LLMs is more challenging due to their sequential token generation and exponentially growing output space [21]. Thus, traditional methods like temperature scaling are less effective for long-form generation. To address this, Band et al. [6] proposed a calibration method targeting probabilities associated with user decisions in LLM-generated guidance but noted limitations in calibrating probabilities in RAG contexts.

To address this issue, we propose the Calibrated Retrieval-Augmented Generation (CalibRAG) framework. CalibRAG allows an LLM using RAG to not only select relevant information to support user decision-making but also provide confidence levels associated with that information by utilizing a forecasting function, ensuring well-calibrated decisions based on the retrieved documents. Here, the forecasting function is the surrogate model that predicts the probability of whether the user's decision based on the guidance provided by RAG will be correct. We empirically validate that our CalibRAG significantly improves calibration performance as well as accuracy, compared to other relevant baselines across several datasets. Our contributions can be summarized as follows:

- We propose the CalibRAG framework, which enables well-calibrated decision-making based on the guidance provided by RAG.
- We construct a new dataset by creating labels that indicate how much decisions made using retrieved documents correctly answer the questions, essential for training the forecasting function.
- We outperform existing uncertainty calibration baselines across various tasks involving RAG context in decision-making scenarios.

## 2 Preliminaries

### 2.1 Decision Calibration of Long Form Generation

As discussed in Sec. 1, since human decision-makers tend to over-rely on the outputs of LLMs during the decision-making process, it is crucial to ensure that the confidence in LLMs' outputs is well-calibrated. To address this problem, Band et al. [6] proposes *decision calibration*, which aims to align the confidence of the model's predicted output with the accuracy of the user's decision based on the model output. This allows the user to make a reliable decision based on the model's confidence. Thus, to achieve this goal, we need to ensure that the model not only generates factual information but also its confidence in the generated responses accurately reflects the likelihood of correctness.

To formalize the problem, we introduce the following notations. Let $x \in \mathcal{X}$ represent the question or task for which a user needs to make a decision (e.g., "Should I take melatonin to help with jet lag after a long flight?"), and let $y \in \mathcal{Y}$ denote the corresponding true answer (e.g., "Yes, if taken at a local bedtime."). Here, $\mathcal{X}$ and $\mathcal{Y}$ are the set of all possible questions and answers, respectively. Given the question $x$, the user provides an open-ended query $q(x)$ (e.g., "Write a paragraph about the effects of melatonin on jet lag.") to an LLM as a prompt to gather information for the decision making about $x$. The LLM, denoted as $\mathcal{M}$, generates a long-form response to the query, i.e., $z \sim$

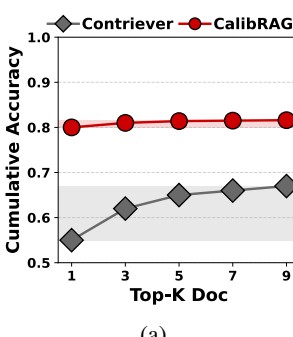 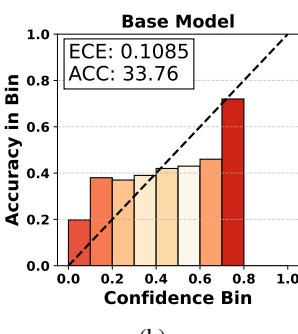 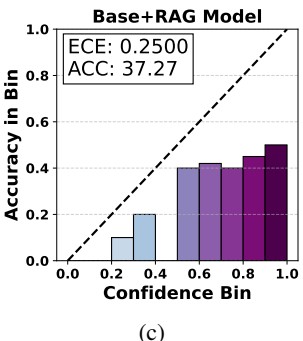

(a)                 (b)                 (c)

Figure 1: **(a) Cumulative accuracy with the top-K documents on our synthetic validation set (see Sec. 3.3).** `contriever-msmarco` gains 11% compared to top-1 when the top-9 documents are used, showing that the top-1 hit is often not optimal. CalibRAG reaches a higher top-1 accuracy and gains little from additional documents. **(b, c) Reliability diagrams on NaturalQA.** For `Llama-3.1-8B` trained under the Number baseline (see Sec. 4), adding the retrieved document (c) raises accuracy relative to the no-document baseline (b) but also increases ECE, indicating greater over-confidence. Bar height is the mean accuracy in each confidence bin; darker shading marks bins with more predictions.

$\mathcal{M}(z|q(x))$, which serves as the guidance for the decision-making process. For the sake of notational simplicity, unless specified otherwise, we will use $q$ in place of $q(x)$. Given the question $x$ and the generated response $z$, the user leverages a forecasting function $f : \mathcal{X} \times \mathcal{Z} \rightarrow \Delta_{|\mathcal{Y}|}$ to assess all possible answers $y \in \mathcal{Y}$, where $\Delta_{|\mathcal{Y}|}$ denotes a simplex over the set $\mathcal{Y}$ and $\mathcal{Z}$ is the space of all possible responses from $\mathcal{M}$. The goal is to use the forecasting function $f$ to ensure that, given the long-form generated LLM response $z$, the user makes calibrated decisions on the question-answer pairs $(x, y)$. Based on this, Band et al. [6] introduces formal definitions for three types of calibrations with varying conditions. For instance, the LLM is *confidence calibrated* [25] with respect to the forecasting function $f$ if $f$ is calibrated on the joint distribution $p(x, y, z)$, that is, for any $\beta \in [0, 1]$

$$\Pr\Big(y = \arg\max_{j \in |\mathcal{Y}|} f(x, z)_j \mid \max_{j \in |\mathcal{Y}|} f(x, z)_j = \beta\Big) = \beta,$$

where $f(x, z)_j$ denotes the $j^{\text{th}}$ element of vector $f(x, z)$.

However, the method proposed by Band et al. [6] to tackle this calibration problem has three major limitations. 1) It requires supervised fine-tuning for three different LLMs, including the LLM responsible for generating a response $z$ and the forecasting function $f$ parameterized with two LLMs. 2) it further needs proximal policy optimization [PPO; 26] for fine-tuning the LLM for response generation, which is known to suffer from training instability [27]. 3) It cannot be directly applied to calibrate the probabilities associated with the user decisions based on the guidance by RAG.

## 2.2 Retrieval Augmented Generation (RAG)

RAG [13] employs Dense Passage Retrieval [DPR; 28] to retrieve relevant documents for question answering. DPR encodes questions and documents independently, enabling precomputation and indexing of document embeddings. At inference time, only the question is embedded and matched against the indexed documents via similarity search. Retrieved documents are then provided as additional context to an LLM, often improving accuracy of answer. Despite its effectiveness, RAG remains vulnerable to retrieval errors. Since retrievers are typically trained in an unsupervised manner [29, 30], their similarity scores do not necessarily reflect the utility of documents for downstream decision-making. As shown in Fig. 1a, the top-ranked document retrieved by Contriever [29] often leads to incorrect predictions, whereas lower-ranked documents may yield better outcomes. Furthermore, incorporating irrelevant documents can mislead the LLM, resulting in overconfident but incorrect answers, as illustrated in Fig. 1c. As an alternative to improve retrieval quality, reranking methods [31, 32] have been proposed to reorder candidates based on relevance signals. However, these approaches are typically optimized for ranking metrics (e.g., MRR, NDCG) rather than the correctness of downstream decisions, and thus do not produce calibrated confidence estimates. We provide a detailed discussion of why reranking methods fail to support decision calibration in Sec. A. Existing RAG methods, including those incorporating reranking, lack mechanisms to assess the confidence of retrieved documents. Addressing this limitation requires not only identifying documents

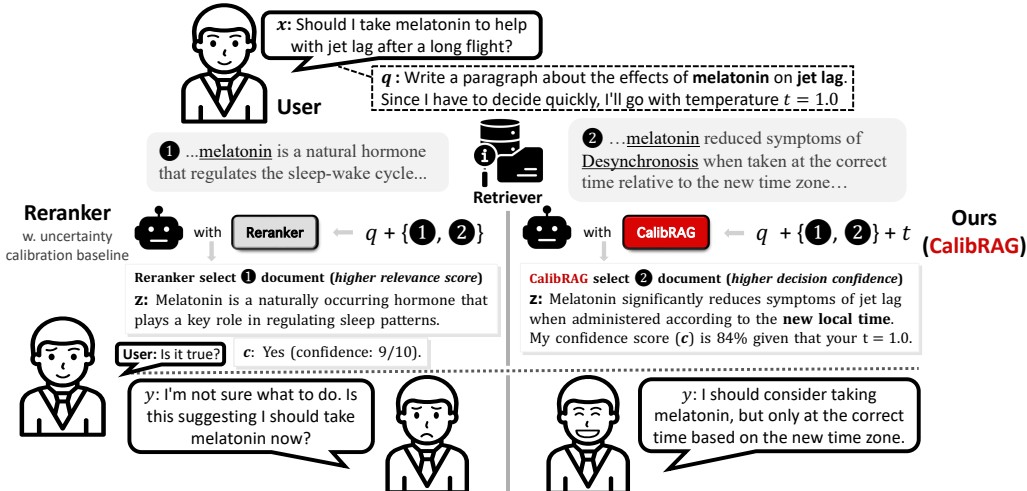

Figure 2: **Comparison between CalibRAG and other reranking methods during inference.** In contrast to conventional methods that rely on relevance scores to rerank retrieved documents, CalibRAG leverages a confidence score derived from the user's risk tolerance $t$ to guide the reranking process.

that better support accurate downstream decisions, but also calibrating the likelihood that such decisions are correct given the retrieved context.

# 3 CalibRAG: RAG for Decision Calibration

**Overview.** When LLMs support user decision-making using RAG, the ability to predict the reliability of those decisions in advance can significantly improve the safety and trustworthiness of the overall experience. Given a task $x$ on which a user must make a decision, we assume an associated open-ended query $q$. The retriever then selects a relevant document $d$ from an external database. Conditioned on $q$ and $d$, the LLM $\mathcal{M}$ produces a long-form guidance $z$ to support the decision. In addition to this guidance, we must also provide a confidence score $c$. This score can be obtained in two ways: (i) by asking $\mathcal{M}$ to verbalize its own confidence, or (ii) by predicting it using a separately trained forecasting function $f$. In our framework, because verbalized confidences from $\mathcal{M}$ are often poorly calibrated, we adopt the latter. The forecasting function outputs $c = f(t, q, d)$, a probability that the user's decision will be correct, conditioned on a temperature parameter $t$ of the decision-making user, $q$, and $d$. While prior work defines $f(x, z)$ over LLM outputs [6], our formulation $f(t, q, d)$ predicts correctness before $z$ is generated. This enables reranking and simplifies supervision. The user ultimately consults $z$ and $c$ to make a decision, and our goal is to ensure that the predicted confidence $c$ is well aligned with the empirical accuracy of decision-making scenarios. Fig. 2 **right** illustrates the overall usage process of CalibRAG.

## 3.1 Problem setup

To train and evaluate the forecasting function $f$, we introduce an LLM-based surrogate user model $U$ that can simulate human decision-making without requiring costly human annotations [6]. We construct prompts such that $U$ makes decisions by referencing both the guidance $z$ and the confidence score $c$, deciding whether to accept the guidance based on the confidence (see Sec. G for prompt details). Since the output of the surrogate user model $U$ is long-form text rather than a class label, we utilize the GPT-4o-mini model (denoted by $\mathcal{G}$) to automatically evaluate whether the user's response matches the correct answer $y$.

To construct supervision signals for $f$, we sample responses from $U$ across various decoding temperatures $t \in \mathcal{T}$ to simulate a range of user behaviors. This temperature reflects behavioral traits of the user $U$, such as risk tolerance or decision urgency. For instance, if the user needs to make a decision quickly and prefers reliability, they can choose a lower $t$. On the other hand, if the user has more time and is open to exploring diverse alternatives, they may opt for a higher $t$. For each combination $(t, q, d)$, we collect $R$ responses and use the proportion of correct ones as a soft label

for training $f$. As a result, the label is represented as a probability between $[0, 1]$, used as a soft label for binary modeling or extended to a multi-class histogram to support finer-grained calibration.

Then, our final goal is to satisfy the following binary calibration condition:

$$\Pr\left[y = U(x, z, f(t, q, d)) \mid f(t, q, d) = \beta\right] = \beta, \quad \forall \beta \in [0, 1], \tag{1}$$

which means that the predicted confidence $\beta$ should match the actual accuracy of user decisions.

As discussed in Sec. 1 and illustrated in Fig. 1c, asking the RAG model to provide its own confidence often leads to excessively high confidence scores, which can cause users to overtrust inaccurate information. In this work, to overcome this limitation, we train a forecasting function $f(t, q, d)$ instead to satisfy the calibration condition in Eq. 1, using supervision signals derived from self-consistency sampling guided by $U$. This approach enables better modeling of real user behavior and allows for more effective confidence calibration, ultimately resulting in a safer and more trustworthy decision-making system.

## 3.2 Modeling and Training

To model the forecasting function $f$, it is essential to have the capacity to sufficiently analyze the impact of the generated guidance $z$ on the actual decision. For this reason, we use a pre-trained LLM $\mathcal{M}$, responsible for generating $z$, as the base feature extractor model $f_{\text{feat}}$. For details of the feature-extraction process, please refer to Sec. B.4. To incorporate the temperature parameter $t$, we apply a Fourier positional encoding [33] that maps the scalar $t \in \mathbb{R}$ to a $2N$-dimensional vector as follows:

$$\text{PE}(t) = \left[\sin(\omega_1 t), \cos(\omega_1 t), \ldots, \sin(\omega_N t), \cos(\omega_N t)\right], \quad \omega_n = 2^n \frac{2\pi}{t_{\max} - t_{\min}}. \tag{2}$$

where $N$ is the number of frequencies in the encoding. The resulting $\text{PE}(t) \in \mathbb{R}^{2N}$ is projected with a learnable matrix $W_p \in \mathbb{R}^{h \times 2N}$ and then added element-wise to the base feature $f_{\text{feat}}(q, d)$. Additionally, to model the probability that $U(x, z, f(t, q, d))$ leads to a correct decision, we place a linear classification head on top of the extractor $f_{\text{feat}}$ derived from the frozen LLM $\mathcal{M}$. For parameter-efficient fine-tuning and to avoid abrupt representation shifts, we keep $\mathcal{M}$ frozen and train only the LoRA adapters [LoRA; 34] and the lightweight head.

The resulting forecasting function is defined as:

$$f(t, q, d) := \sigma\left(W_{\text{head}}^{\top}\left(f_{\text{feat}}(\text{concat}[q, d]; W_{\text{LoRA}}) + W_p \, \text{PE}(t)\right) + b_{\text{head}}\right), \tag{3}$$

where $\sigma(x) = 1/(1 + \exp(-x))$ is the sigmoid function. Here, $W_{\text{head}}$, $b_{\text{head}}$, $W_p$, and the LoRA parameters $W_{\text{LoRA}}$ are all learnable. This formulation allows $f$ to condition its prediction on both the semantic compatibility of the $q$-$d$ pair and the user-specific behavior encoded by $t$, providing an uncertainty-aware estimate of decision correctness. Then we train $f$ by minimizing the following log-likelihood loss:

$$\mathcal{L} = -\frac{1}{|\mathcal{S}|} \sum_{(t, q, d, b) \in \mathcal{S}} \left[b \log f(t, q, d) + (1 - b) \log(1 - f(t, q, d))\right] \tag{4}$$

where $\mathcal{S}$ is the training dataset and $b \in [0, 1]$ is the soft label derived from self-consistency sampling. This objective encourages the predicted confidence $f(t, q, d)$ to match the empirical correctness probability, enabling the model to generalize across varying decision behaviors encoded by $t$, while leveraging the LLM's latent representations for calibrated prediction. Although the primary formulation uses soft binary labels, we also investigate a multi-class variant where the correctness distribution is discretized into histogram bins and train $f$ using a categorical objective. Full experimental results and comparisons are provided in Sec. 4.

*Remark.* Among various scoring rules [35, 36, 6] used to measure the forecast quality of functions predicting the true probability $p$, the *strictly proper scoring rule* has the advantage that its *unique maximizer* is the true probability $p$. Consequently, training a forecast function using a strictly proper scoring rule as the training objective ensures that the forecasts are learned to be as close as possible to the true probability $p$ as the number of training examples increases. Note that the loss in Equation 4 to train our forecast function $f$ is an example of a strictly proper scoring rule, the logarithmic score. This makes our loss function crucial for training $f$ to produce well-calibrated predictions.

### 3.3 Synthetic Supervision Data Generation

To conduct the supervised learning discussed in Sec. 3.2, it is essential to construct an appropriate synthetic training dataset $\mathcal{S}$ consisting of the triples $(t, q, d, b)$. We first extract the $(x, y)$ decision-making task pairs from the following three Question Answering datasets: 1) TriviaQA [37], 2) SQuAD2.0 [38], and 3) WikiQA [39] datasets. Then, for every $x$ in the training dataset, we generate an open-ended query $q$ based on each $x$, using the GPT-4o-mini model. At this point, it is important to note that instead of retrieving only the single top document $d$ with the highest similarity score from the retriever model for each query $q$, we retrieve the top 20 documents. There are two reasons for this. First, as illustrated in Fig. 1a, a large number of low-ranked documents actually help the surrogate user make a correct decision. If we only include the top-1 documents, many of which would be labeled as incorrect, the synthetic dataset would be highly biased to negative samples. Second, using only one $d$ per $(x, y)$ pair for labeling and training could result in the model overfitting to the label without learning the relationship between $q$ and $d$ adequately. By pairing the same $q$ with various $d$'s, the model can learn from positive and negative samples, improving its ability to generalize. After retrieving multiple documents, we provide each $(q, d)$ pair to the RAG model $\mathcal{M}$, which generates the guidance $z$ based on $d$.

Then, the user model $U$ receives the task $x$ and guidance $z$, and samples responses with the decoding temperature $t$, which reflects behavioral variation during sampling. To estimate the reliability of the generated decision, we sample $R = 10$ from $U$ at the same temperature and evaluate each using the correctness function $\mathcal{G}$, which compares the decision with the ground truth answer $y$. The soft label $b \in [0, 1]$ is then computed as the proportion of correct responses among the $R$ samples. Thus, for each $(x, y)$ pair, we generate multiple training quadruples $(t, q, d, b)$, each corresponding to a different $t$ setting and document retrieved. Refer to Sec. E for examples of our synthetic data.

### 3.4 Inference

After finishing the training of the forecasting function $f$, we perform inference for a new decision task $x^*$ through the following four stage process:

**Stage 1: Initial retrieval of documents**. Given an open-ended query $q^*$, derived from the original question $x^*$, we begin the document retrieval process using the retrieval model. Similarly to the training data generation process, we retrieve the top $K$ relevant documents from the external database, denoted by $\mathcal{D}^* := \{d_i^*\}_{i=1}^K$. The goal of this stage is to construct a diverse set of candidate documents that may contain valuable information for producing the correct answer $y$.

**Stage 2: Scoring and selection of documents.** Once the $K$ candidate documents are retrieved, we estimate the decision confidence of each document with our trained $t$-conditioned forecasting function $f$. At inference time, the user may choose $t$ to reflect their decision preference, with lower values for cautious, consistent decisions and higher values for exploratory reasoning. Regardless of the original retrieval similarity score, each document is then reranked by its predicted confidence. Concretely, we sort the documents in descending order of the probabilities $\{f(t, q^*, d_i^*)\}_{i=1}^K$, which represent the chance that the user will reach a correct decision if guidance $z$ is generated from each document. The top-ranked document is selected for the next stage. Here, if the predicted probability for the highest-ranked document $d^*$ falls below a predefined threshold $\epsilon$ (defaulted to 0.5, with further details provided in § D.1), we consider the guidance $z$ insufficient for a reliable decision at temperature $t$. In this case, we move on to Stage 3, where we retrieve a new set of $K$ candidate documents to search for documents that provide higher confidence. If this condition is not met, we move forward to Stage 4.

**Stage 3: Reformulating the query (Optional).** If the predicted probability for the highest-ranked document $d^*$ is lower than a pre-defined threshold $\epsilon$ in Stage 2, to retrieve a new set of $K$ candidate documents, we reformulate our open-ended query $q^*$ into $q^{**}$ by emphasizing more important content from the question $x$. This reformulation focuses on extracting key aspects of the original task, ensuring that the next retrieval attempt targets more relevant and helpful documents. After reformulating the query, we repeat Stage 1 and Stage 2 once again. Examples of query reformulation are shown in § C.

**Stage 4: Final decision.** After retrieving the document $d^*$, we generate the guidance $z^*$ using the RAG model $\mathcal{M}$. Then, the user model $U$ makes a decision $U(x^*, z^*, f(q^*, d^*, t))$, with the

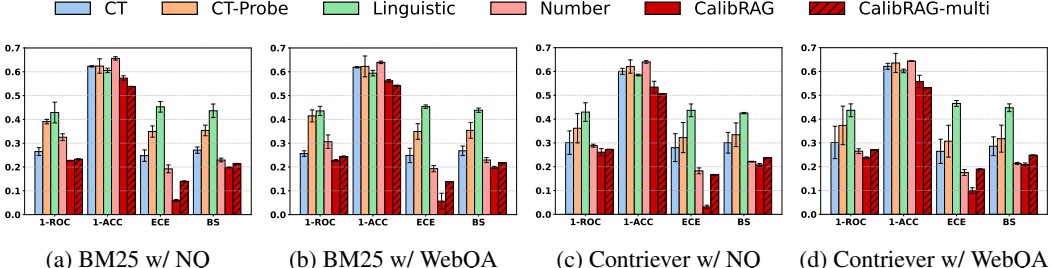

Figure 3: **Evaluation results of the baselines and CalibRAG using two retriever models: BM25 and Contriever on NQ and WebQA.** We report four metrics—1-AUROC, 1-ACC, ECE, and BS—where *lower values indicate better performance.*

decoding temperature $t$ selected in Stage 2. This decision is compared with the correct answer $y^*$ by $\mathcal{G}$ to determine its accuracy.

# 4 Experiments

**Implementation detail.** For all experiments, following Sec. 3.3, we collect a total of 20,870 samples for training and 4,125 for validation. All evaluations are conducted in a **zero-shot** setting on held-out tasks that are disjoint from both the training and validation sets. Unless otherwise specified, we use `Llama-3.1-8B` [3] as both the RAG model $\mathcal{M}$ and decision model $U$. However, we also conduct ablation studies diverse model structures, and the results are presented in Sec. D. To evaluate long-form generated answers, we employ `GPT-4o-mini` as the evaluation model $\mathcal{G}$.

**Baselines.** We compare CalibRAG with the following relevant baselines.

- **Uncertainty calibration baselines:** (1) *Calibration Tuning* [21] methods employ a model that outputs probabilities for answering "Yes" or "No" to the question, "Is the proposed answer true?" These probabilities allow us to measure the uncertainty calibration of the response. As baselines, we consider two variants: **CT-probe**, which adds a classifier head to estimate the probability of correctness, and **CT-LoRA**, which utilizes the normalized token probability between the tokens "Yes" and "No." (2) *Verbalized Confidence Fine-tuning* [40, 41, 6] utilizes verbalized tokens to represent the model's confidence. In this case, we also consider two baseline variants: **Number-LoRA**, which expresses confidence as an integer between 0 and 10, and **Linguistic-LoRA**, which uses linguistic terms (e.g., "Doubtful" or "Likely") to indicate confidence. For all uncertainty calibration baselines, guidance and confidence are generated based on the top-1 document retrieved by the retriever.

- **Reranking and Robust RAG baselines:** Although CalibRAG is primarily designed to enable well-calibrated decision-making in RAG-guided scenarios, it can also be interpreted as a reranking approach for retrieved documents in downstream tasks as a consequence of **Stage 2** during inference. Accordingly, we compare CalibRAG against two reranking baselines and one robust RAG baseline: (1) **Cross-encoder** with MiniLM-L6-v2, which reranks documents based on the similarity score of the jointly embedded query and documents with cross attention. (2) **LLM-rerank** [42], which prompts the LLM, `GPT-3.5-turbo`, to rerank documents by leveraging the relationship between the query $q$ and the document $d$. (3) **SelfRAG** [43], a robust RAG baseline that dynamically determines the necessity of retrieval and self-evaluates the relevance of a retrieved document $d$ to the query $q$, as well as the usefulness of the generated guidance $z$ for $q$, using special tokens such as "Retrieve", "IsREL", "IsSup", and "ISUSE".

**CalibRAG.** Unless otherwise specified, CalibRAG reranks the top-20 documents at inference. The forecasting function is evaluated by marginalizing over a set of six decoding temperatures $t \in \{1.0, 1.1, \ldots, 1.5\}$. This is because, in the absence of explicit information about a user's preference, the forecasting function cannot accurately model the user's behavior under a single decoding temperature. To reflect this variation, we approximate marginalization by averaging over these six values, as exact integration over all possible $t$ is infeasible. Building on this, CalibRAG-multi extends CalibRAG to a multi-class setting by modeling the correctness histogram across bins (0-10).

**Evaluation metrics.** We evaluate all the models in terms of accuracy, AUROC, and various calibration metrics such as Expected Calibration Error [ECE; 44], Brier Score [BS; 45]. For clarity and

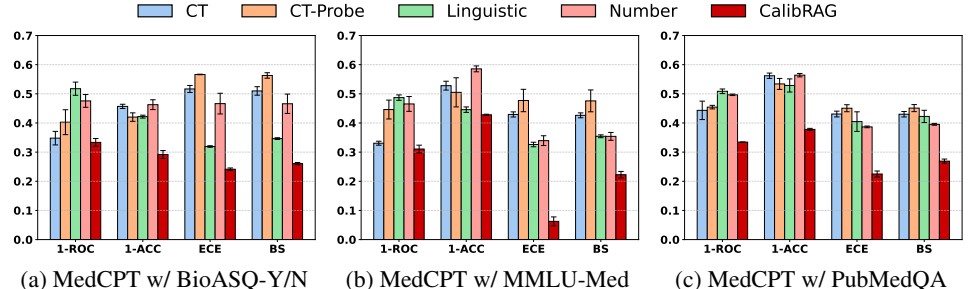

| | CT | CT-Probe | Linguistic | Number | CalibRAG |

(a) MedCPT w/ BioASQ-Y/N      (b) MedCPT w/ MMLU-Med      (c) MedCPT w/ PubMedQA

Figure 4: **Evaluation results of the baselines and CalibRAG utilizing MedCPT on BioASQ-Y/N, MMLU-Med, and PubMedQA.** We report four metrics—1-AUROC, 1-ACC, ECE, and Brier Score—where *lower values indicate better performance.*

Table 1: Comparison of RAG variants across datasets.

(a) Comparison of reranking RAG methods. Baseline reranking scores are treated as confidence.

| Data | Method | AUROC (↑) | ACC (↑) | ECE (↓) | BS (↓) |
|------|--------|-----------|---------|---------|--------|
| HotpotQA | Cross-encoder | 60.74 | 34.98 | 0.477 | 0.477 |
| | LLM-rerank | 60.57 | 38.52 | 0.248 | 0.297 |
| | CalibRAG | **72.47** | **42.37** | **0.106** | **0.206** |

(b) Comparison of robust RAG methods using Llama-2-7B as $\mathcal{M}$.

| Data | Method | AUROC (↑) | ACC (↑) | ECE (↓) | BS (↓) |
|------|--------|-----------|---------|---------|--------|
| NQ | SelfRAG | 48.4 | 36.2 | 0.522 | 0.545 |
| | CalibRAG | **63.5** | **37.4** | **0.258** | **0.287** |
| WebQA | SelfRAG | 51.9 | 39.7 | 0.478 | 0.503 |
| | CalibRAG | **68.8** | **40.5** | **0.217** | **0.262** |

consistency, we adopt the 1-AUROC and 1-ACC notation in plots so that all metrics can be interpreted under the convention that lower values indicate better performance. Details regarding these metrics can be found in Sec. B.3.1.

## 4.1 Main Results

**Comparison with uncertainty calibartion baselines.** Fig. 3 and Fig. 4 present a comparison of uncertainty-based baselines in both general and specific domain tasks.

For the general domain, we evaluated CalibRAG on the Natural QA (NQ) [46] and WebQA [47] datasets. To demonstrate that our method performs well across different retrievers, we conducted experiments using both Contriever [29] and BM25 [48] retrievers. The results in Fig. 3 clearly show that CalibRAG and CalibRAG-multi outperform other baselines in all metrics. In particular, it significantly reduces calibration metrics across all datasets and retriever settings, indicating that CalibRAG effectively calibrates the decision-making process compared to other baselines. Additionally, while CalibRAG is not explicitly designed to improve accuracy, it naturally identifies documents that are more likely to lead to correct decisions. As a result, it also enhances accuracy compared to other baselines. We also evaluate CalibRAG without reranking, which still shows improved calibration metrics compared to both the baselines and the setting where baseline confidence scores are used for reranking. Detailed results can be found in Sec. D.

For domain specific evaluation, we assess CalibRAG on the BioASQ-Y/N, MMLU-Med, and PubMedQA datasets from the MIRAGE benchmark [49], which focus on the medical domain. Since domain-specific retrievers are necessary in this field, we utilize MedCPT [50], a retriever trained on user click logs on PubMed corpus. Note that the rest of CalibRAG model, including the LLM $\mathcal{M}$ and the forecasting function, which have been trained on TriviaQA, SQuAD2.0, and WikiQA, remains fixed. Thus, this setup evaluates robustness of CalibRAG when applied to an unseen retriever and out-of-domain datasets. The results in Fig. 3 clearly demonstrate that CalibRAG outperform other baselines across all metrics, even in specialized domain scenarios. Similar to the general domain, CalibRAG significantly improves calibration metrics across all datasets, showing its ability to effectively calibrate the decision-making process even with an unseen retriever and dataset.

**Comparison with reranking and robust RAG baselines.** CalibRAG retrieves the top $K$ documents during inference and selects the one most likely to lead to a correct decision, using it to generate guidance and confidence for a well-calibrated decision-making process. To further evaluate its effectiveness, we compare CalibRAG against reranking and robust RAG baselines in Table 1a and Table 1b.

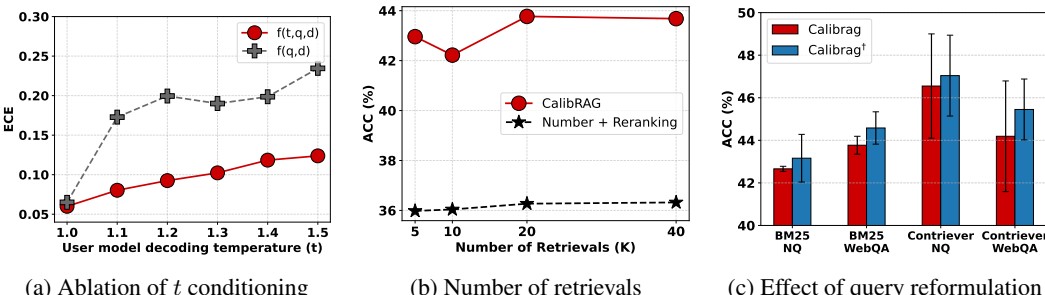

| (a) Ablation of $t$ conditioning | (b) Number of retrievals | (c) Effect of query reformulation |

Figure 5: **(a)** Calibration with and without temperature conditioning on the NQ dataset using Contriever. **(b)** Effect of the number of retrieved documents on reranking performance on the WebQA dataset using BM25. **(c)** Impact of query reformulation during inference.

First, we conduct a comparison with reranking baselines using the HotpotQA [51] dataset. To assess the calibration performance of existing reranking baselines, we use their reranking scores as confidence values. Surprisingly, Table 1a shows that CalibRAG not only improves calibration performance compared to the baselines but also enhances accuracy. This result suggests that, while our inference procedure is designed to identify documents that lead to correct decisions, it also effectively retrieves documents that are highly relevant to the query $x$. The confidence assignment method for reranking models and the comparison with the BEIR reranking benchmark can be found in Sec. D.2.

Next, we compare CalibRAG against a robust RAG baseline using the NQ and WebQA datasets. To evaluate the calibration performance of SelfRAG, we use the "Utility" score tokens as the confidence score. Similar to the reranking baseline comparison, Table 1b demonstrates that CalibRAG consistently outperforms other baselines across all datasets and metrics.

### 4.2 Ablation Studies

**Does temperature conditioning improve calibration?** To evaluate the impact of temperature-aware modeling, we compare our full model $f(t, q, d)$ with a temperature-agnostic variant $f(q, d)$ trained without conditioning on user decoding behavior. As shown in Fig. 5a, temperature-aware forecasting significantly improves ECE across user model temperatures. In particular, $f(q, d)$ tends to overestimate confidence under higher-temperature sampling, leading to increased ECE. In contrast, our proposed $f(t, q, d)$ correctly adapts to user-specific (decoding) behaviors and maintains relatively low ECE across the full temperature range. This confirms that incorporating user variability via temperature conditioning enables more reliable decision calibration.

**How does the number of retrieved passages ($K$) impact reranking?** We use $K = 20$ documents for reranking in all experiments, balancing the computational cost and the performance of the decision-making task. To validate our choice, we plot accuracy as a function of the number of documents for reranking in Fig. 5b. The results show that performance improves up to 20 documents, but the gains diminish beyond 40 documents, supporting our choice of 20 documents.

**Effect of Query Reformulation** During inference, if none of the $K$ retrieved documents in **Stage 2** are predicted to contribute to a correct decision, CalibRAG can optionally proceed to **Stage 3**, where it reformulates the query $q$ and retrieves a new set of $K$ documents. To evaluate whether this optional step improves performance, we conducted experiments on the NQ and WebQA datasets using both BM25 and Contriever retrievers. As shown in Fig. 5c, incorporating **Stage 3**, denoted as CalibRAG$^\dagger$, consistently improves performance across all cases. However, this improvement comes at the cost of additional computation, as reformulating the query and retrieving new documents require extra processing.

## 5 Conclusion

In this paper, we introduced CalibRAG, a simple yet effective framework designed to ensure that the RAG-guided decision-making process is well-calibrated. Our experiments demonstrated that CalibRAG significantly enhances QA performance within the RAG setting across various datasets and retriever models. Moreover, ablation studies showed that CalibRAG effectively aligns model

confidence with factual correctness, resulting in improved decision-making accuracy and calibration. Overall, CalibRAG stood out as a robust solution for enhancing the reliability of RAG-based LLM guidance in decision-driven scenarios. However, creating synthetic datasets and training the forecasting function for decision calibration may introduce some overhead. Nonetheless, accurately calibrating language model confidence is crucial, making this approach both valid and worthwhile.

## Acknowledgement

This work was supported by Institute of Information & communications Technology Planning & Evaluation(IITP) grant funded by the Korea government(MSIT) (No.RS-2019-II190075, Artificial Intelligence Graduate School Program(KAIST)). This work was supported by Institute of Information & communications Technology Planning & Evaluation(IITP) grant funded by the Korea government(MSIT) (No.RS-2024-00509279, Global AI Frontier Lab). This work was supported by the National Research Foundation of Korea(NRF) grant funded by the Korea government(MSIT) (NRF-2022R1A5A708390812). This work was supported by Institute of Information & communications Technology Planning & Evaluation(IITP) grant funded by the Korea government(MSIT) (No.2022-0-00184, Development and Study of AI Technologies to Inexpensively Conform to Evolving Policy on Ethics). This work was supported by the Institute of Information & Communications Technology Planning & Evaluation(IITP) grant funded by the Korea government(MSIT) (No.RS-2025-02219317, AI Star Fellowship(Kookmin University)). This work was supported by Artificial intelligence industrial convergence cluster development project funded by the Ministry of Science and ICT(MSIT, Korea) & Gwangju Metropolitan City. This research was supported by the MSIT(Ministry of Science, ICT), Korea, under the Global Research Support Program in the Digital Field program(IITP-2024-RS-2024-00417958) supervised by the IITP(Institute for Information & Communications Technology Planning & Evaluation).

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

# A  Related Works

## A.1  Uncertainty Calibration in Language Models

Traditional calibration methods rely on token-level log probabilities [25], but modern LLMs generate text autoregressively by multiplying conditional probabilities [4]. Estimating semantic-level probabilities would require marginalization over all possible sequences, which is computationally intractable. As a result, token-level probabilities often fail to provide reliable confidence estimates for long-form text generation.

Prompt-based approaches aim to address this problem by eliciting verbalized confidence scores [40, 52]. For example, a model can be prompted with: *"Express your confidence as a number between 0 and 100."* If it responds with *"90"*, this value is interpreted as its confidence level. However, LLMs often exhibit overconfidence in zero-shot settings, resulting in poorly calibrated outputs [9]. Although RAG can mitigate this issue, when the retrieved context is unreliable, LLM may still demonstrate overconfidence, leading to misleading conclusions. Addressing this challenge remains essential for improving LLM reliability in complex decision-making tasks.

## A.2  Methods for Enhancing RAG Robustness

Recent advancements in reranking for RAG have largely focused on enhancing the relevance of retrieved documents with respect to the input query. For example, LLM-based rerankers leverage semantic representations to reorder documents based on their relevance [42], while cross-encoder-based rerankers jointly encode query-document pairs to model their interaction more precisely [53]. These approaches are highly effective in improving retrieval relevance and downstream QA performance. However, they are fundamentally designed to rank documents by relevance, not to assess how the retrieved information influences the correctness of the final user decision based on the LLM-generated answer. Thus, the resulting scores, although often normalized between 0 and 1, are not calibrated probabilities of correctness and cannot be directly used for decision calibration.

Similarly, Self-RAG [43] introduces the notion of utility scores for retrieved documents to identify potentially helpful content. While this provides a signal for filtering noisy documents, the utility score reflects plausibility rather than empirical correctness. As such, these scores are neither optimized for nor aligned with standard calibration metrics such as ECE, NLL, or Brier Score.

In contrast, our approach directly addresses this gap by training a forecasting function to output calibrated confidence scores that reflect the actual correctness of decisions made by a surrogate user model. We explicitly supervise the forecasting function using binary labels that indicate whether the model's final prediction is correct, and optimize this function using strictly proper scoring rules. This ensures that the predicted confidence scores match the empirical likelihood of correctness, thus enabling true decision calibration rather than merely relevance estimation.

This fundamental difference in supervision **signal** (relevance vs. correctness) and **objective** (ranking vs. calibration) delineates the core novelty of our work from prior reranking-based approaches. By aligning the model's confidence estimates with empirical decision accuracy, our method offers a principled and interpretable framework for improving trustworthiness in RAG systems.

# B  Experimental details

Our implementation builds on key libraries such as PyTorch 2.1.2 [54], Hugging Face Transformers 4.45.1 [55], and PEFT 0.7.1,[3] providing a robust foundation for experimentation. We employ the `Llama-3.1-8B-Instruct` model, an open-source multilingual LLM available on Hugging Face.[4] Our experiments are conducted on NVIDIA RTX 3090 and RTX A6000 GPUs. Additionally, we utilize the official `facebookresearch-contriever` repository[5] and the `elastic-research-bm25` repository[6] for our retrieval model. We also use `MedMCT`

---

[3] https://github.com/huggingface/peft
[4] https://huggingface.co/meta-llama/Meta-Llama-3.1-8B-Instruct
[5] https://github.com/facebookresearch/contriever
[6] https://www.elastic.co/

based on the MedRAG framework.[7] For training calibration tuning baselines, we reference the `calibration-tuning` repository.[8]

### B.1 Datatsets

**Train Datasets** SQuAD [56, 38] is a reading comprehension dataset sourced from Wikipedia, containing questions answered by text spans from the articles. WikiQA [39] is a question-sentence pair dataset from Wikipedia, designed for open-domain question answering and includes unanswerable questions for research on answer triggering. TriviaQA [37] is a reading comprehension dataset with questions authored by trivia enthusiasts, paired with evidence documents from Wikipedia and other web sources. We randomly sampled 10,000 data points each from TriviaQA and SQuAD2.0, and collected all 873 training samples from WikiQA. In addition, we incorporated non-overlapping samples from SQuAD1.0, resulting in a combined training dataset of 61,886 examples after deduplication. For the validation set, we gathered 2,000 samples each from TriviaQA and SQuAD, along with 126 samples from WikiQA, and added non-overlapping samples from SQuAD1.0, yielding a total of 12,643 validation data points. All null values were removed prior to finalization. We downloaded all these datasets in Hugging Face datasets [9].

For the construction of the labeled dataset $\mathcal{S}$ used to train the forecasting function of CalibRAG, we sample a temperature $t \sim \mathrm{Uniform}[1.0, 2.0]$ for each query $q$ and retrieved document $d$. For each triplet $(t, q, d)$, we perform user decoding 10 times and assign a soft label $b$ indicating the ratio of generated answers that contain the ground truth. The final dataset $\mathcal{S}$ thus consists of tuples in the form $(t, q, d, b)$. **The dataset will be made publicly available upon the publication of this work.**

**Evaluation Datasets** For zero-shot evaluation, we employ several datasets covering diverse domains and question types. HotpotQA [51] is a multi-hop question-answering dataset requiring reasoning across multiple supporting documents from Wikipedia to find answers, emphasizing a more complex retrieval and reasoning process. WebQA [47] is an open-domain question-answering dataset consisting of natural, conversational questions paired with web documents, targeting real-world, context-rich scenarios. Natural Questions (NQ) [46] is another large-scale question-answering dataset, designed to answer questions based on Wikipedia articles, containing both long-form and short-form answers. These datasets are used without additional training, providing a robust evaluation of the generalization capabilities of CalibRAG across different domains and question types.

We also evaluate domain-specific datasets, including BioASQ [57], a biomedical QA dataset containing factoid, list, and yes/no questions derived from PubMed articles, as well as Medical Information Retrieval-Augmented Generation Evaluation (MIRAGE) [49] and a textbook corpus.

### B.2 Hyperparameters

Table 2 outlines the hyperparameters used for training the base model and LoRA, including key parameters such as learning rate, batch size, and LoRA-specific settings like rank and alpha.

### B.3 Evaluation metrics

To evaluate long-form text, we utilized `gpt-4o-mini` to compare the ground-truth answers with the predicted answers in all cases. Based on this comparison, we labeled each instance as correct or incorrect accordingly.

#### B.3.1 Calibration metrics

- **Expected Calibration Error** [ECE; 44]:

$$\mathrm{ECE} = \sum_{m=1}^{M} \frac{|B_m|}{n} \left| \mathrm{acc}(B_m) - \mathrm{conf}(B_m) \right|$$

---

[7] https://github.com/Teddy-XiongGZ/MedRAG
[8] https://github.com/activatedgeek/calibration-tuning
[9] https://github.com/huggingface/datasets

Table 2: Hyperparameters for LLM Training

| Base Model Hyperparameters | | LoRA Hyperparameters | |
|---|---|---|---|
| **Hyperparameter** | **Value** | **Hyperparameter** | **Value** |
| Learning Rate | $\{10^{-4}, 10^{-5}\}$ | LoRA Rank | 8 |
| Batch Size | $\{1, 4\}$ | LoRA Alpha | 16 |
| Max Steps | 10,000 | LoRA Dropout | 0.1 |
| Optimizer | AdamW | | |
| Dropout Rate | 0.0 | | |
| Gradient Accumulation Steps | [1, 4] | | |
| Weight Decay | 0.01 | | |
| Gradient Clipping | 1.0 | | |
| Warmup Steps | 500 | | |
| Scheduler | Linear | | |

where $B_m$ is the set of predictions in bin $m$, $\text{acc}(B_m)$ is the accuracy, and $\text{conf}(B_m)$ is the average confidence of the predictions in that bin. ECE measures how well the model's predicted probabilities are calibrated.

- **Brier Score** [BS; 45]:

$$\text{BS} = \frac{1}{N} \sum_{i=1}^{N} (f_i - y_i)^2$$

where $f_i$ is the predicted probability and $y_i$ is the true label. BS combines both the accuracy and confidence of the predictions, penalizing overconfident and underconfident predictions.

- **Negative Log Likelihood (NLL)**:

$$\text{NLL} = -\frac{1}{N} \sum_{i=1}^{N} \log p(y_i \mid x_i)$$

where $p(y_i \mid x_i)$ is the probability assigned to the correct class $y_i$ given input $x_i$. NLL evaluates the model's probabilistic predictions and lower values indicate better calibration.

## B.4 CalibRAG Details

**Feature extraction details.** To extract features for the forecasting function, we use the hidden state of the last token from the second-to-last layer of the LLM $\mathcal{M}$, as it empirically yielded better calibration performance than other layers. This hidden state serves as the input representation $f_{\text{feat}}(q, d)$ for the classifier.

**Positional encoding details.** We use a Fourier positional encoding with $N = 6$ frequency components to encode the temperature parameter $t$. This encoding covers the range $t \in [1.0, 2.0]$, and during training data construction, we sample $t$ uniformly from this range to simulate diverse user behaviors.

## C  Examples of query reformulations

In CalibRAG, the initial query is generated to simulate how a human decision-maker might pose a simple query based on the input. For example, a decision-maker faced with a problem such as "Is a tomato a fruit or a vegetable?" might craft a straightforward query like "Classification of tomatoes" to query a language model. Using this setup, we employed an LLM generator to create simple yet relevant queries and retrieved documents based on these queries. If the retrieved documents were insufficiently informative, the query was reformulated in Stage 3. This reformulation emphasized key terms to refine the query and improve the quality of retrieved documents. The specific prompt used for this process is detailed in § G.

Table 3: Examples of Query Reformulation

| Case | Original Query | Reformulated Query |
|------|----------------|--------------------|
| 1 | Write a paragraph about the effect of TRH on myocardial contractility. | Write a paragraph about the effect of Thyrotropin-Releasing Hormone (TRH) on myocardial contractility. |
| 2 | Write a paragraph about the clinical trials for off-label drugs in neonates as cited in the literature. | Write a paragraph about clinical trials for off-label drug use in neonates as reported in the medical literature. |
| 3 | Write a paragraph about the current representatives from Colorado. | Write a paragraph about the current representatives from the state of "Colorado" in the United States. |
| 4 | Write a paragraph about the current minister of local government in Zimbabwe and their role within the government. | Write a paragraph about the current Minister of Local Government and Public Works in Zimbabwe and their role within the government. |

Table 4: Effect of Threshold Selection on Performance. Experiments on the BioASQ dataset show how increasing $\epsilon$ affects accuracy and calibration metrics.

| $\epsilon$ | AUROC | ACC | ECE | BS |
|------|-------|-----|-----|-----|
| 0.0 | $71.21 \pm 0.83$ | $35.03 \pm 0.14$ | $\mathbf{0.2500} \pm 0.01$ | $0.2900 \pm 0.01$ |
| 0.4 | $76.15 \pm 1.50$ | $35.05 \pm 0.25$ | $0.2608 \pm 0.00$ | $0.2830 \pm 0.00$ |
| 0.5 | $76.50 \pm 4.98$ | $35.98 \pm 0.38$ | $0.2667 \pm 0.00$ | $\mathbf{0.2779} \pm 0.01$ |
| 0.6 | $\mathbf{77.20} \pm 4.10$ | $\mathbf{36.50} \pm 0.45$ | $0.2707 \pm 0.00$ | $0.2800 \pm 0.01$ |

To help readers understand the transformation from the initial query to its reformulated version, Table 3 provides examples that illustrate how queries evolve during the refinement process, offering practical insights into the mechanism.

# D   Additional experiments

## D.1   Analysis of $\epsilon$

In our experiments, $\epsilon$ was set as a balanced choice to manage the trade-off between accuracy and calibration error. As shown in Table 4, increasing $\epsilon$ results in retrieving a larger number of new queries, incorporating more relevant information, and thereby improving accuracy. However, this increase can potentially lead to higher calibration errors. Specifically, while better retrieval enhanced prediction accuracy, the confidence scores for these predictions only increased marginally. This mismatch between improved accuracy and relatively low confidence resulted in underconfident predictions, which contributed to a slight increase in calibration error.

To assess the impact of different $\epsilon$ values on model performance, we conducted experiments on the BioASQ dataset. Based on these observations, we selected $\epsilon = 0.5$ as a reasonable compromise to balance accuracy improvements with calibration reliability.

## D.2   Evaluation on BEIR Benchmark

To provide a more comprehensive evaluation, we conducted experiments using two datasets from the BEIR benchmark: SciFact and TREC-COVID. These evaluations aim to validate the effectiveness of CalibRAG beyond its primary focus on well-calibrated decision-making, which predicts the probability of a correct decision when a user relies on the generated guidance to solve a given problem. While CalibRAG is not specifically designed as a reranking method to optimize retrieval performance, it inherently supports both calibration and retrieval.

Table 5: Evaluation results on TREC-COVID and SciFact datasets, a subset of the BEIR benchmark. The evaluation metric is Normalized Discounted Cumulative Gain (NDCG@K).

| Model | Dataset | NDCG@5 | NDCG@10 |
|---|---|---|---|
| Cross-Encoder | TREC-COVID | 0.7655 | 0.7576 |
| | SciFact | 0.6668 | 0.6914 |
| CalibRAG | TREC-COVID | **0.7863** | **0.7660** |
| | SciFact | **0.6872** | **0.7114** |

Table 6: Results of Verbalized Confidence Fine-Tune Evaluation on the MMLU Dataset using `Llama-3.1-8B-Instruct`. Evaluation metrics are ACC and ECE.

| Case | ACC | ECE |
|---|---|---|
| Continuous-Number | 43.63 | 0.3190 |
| Discrete-Number | 44.96 | 0.1605 |
| Linguistic | 45.03 | 0.1585 |

For the experiments, we followed the standard retrieval pipeline, retrieving documents using BM25 and reranking the top-100 results. We compared CalibRAG with the Cross-Encoder baseline, and the results, presented in Table 5, demonstrate that CalibRAG consistently outperforms the Cross-Encoder. These findings validate that CalibRAG not only enables well-calibrated decision-making but also enhances retrieval performance, reinforcing its utility in relevant scenarios.

## D.3 Analysis of Verbalized Confidence Representations

CalibRAG does not rely on linguistic or numerical confidence in its primary approach. Instead, it provides confidence scores based on the probability predictions generated by the forecasting function. Verbalized confidence, however, was used as a baseline in comparative models. Verbalized confidence is typically expressed as a continuous number within the range [0, 100] Tian et al. [40] and Xiong et al. [52], but LLMs often struggle to interpret these numerical values precisely.

To address this limitation, alternative representations were explored in the baselines: (1) linguistic expressions (e.g., "likely"), and (2) discrete numerical values ranging from 0 to 10. These approaches were termed Linguistic and Number, respectively, with detailed prompt designs provided in Appendix E.

To further analyze verbalized confidence, we conducted experiments on the MMLU dataset using the Llama-3-8B model. We evaluated the effectiveness of three confidence representations: continuous number, discrete number, and linguistic. As shown in Table 6, both discrete number and linguistic representations outperformed the continuous number baseline. Linguistic confidence, in particular, addressed the limitations of the model's understanding of numerical relationships and improved calibration.

## D.4 Ablation on user model $U$

We additionally conduct an ablation evaluation on various user models $U$, considering that human users may make different decisions depending on their knowledge background in real-world scenarios. We evaluated the performance of CalibRAG and baseline methods on the NQ and WebQA datasets using two retriever models, BM25 and Contriever. For this, we compare the performance of `Phi-4` [58] and `DeepSeek-Distill` [59], which represent state-of-the-art user models.

As shown in Fig. 6 and Fig. 7, our results demonstrate that CalibRAG consistently achieves better accuracy and calibration error across different user models compared to other baselines.

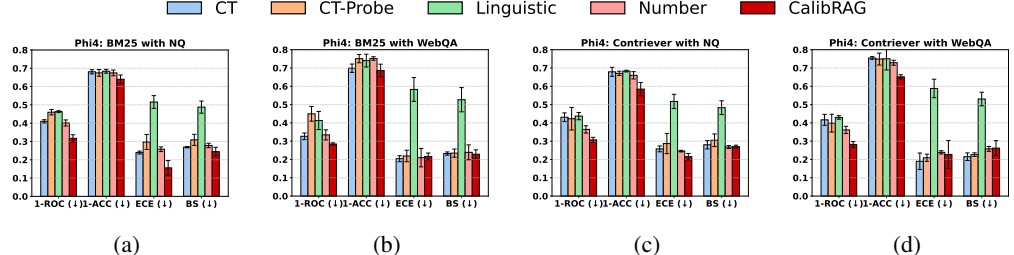

(a)        (b)        (c)        (d)

Figure 6: Evaluation results of the baselines and CalibRAG utilizing two retriever models: BM25 (a, b) and Contriever (c, d) on NQ (a, c) and WebQA (b, d). Here, we utilize `Phi-4` [58] as our user model $U$. We report four metrics—1-AUROC, 1-ACC, ECE, and Brier Score—where lower values indicate better performance.

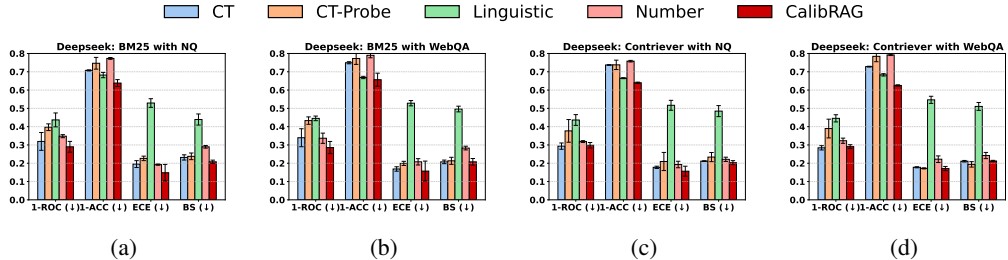

(a)        (b)        (c)        (d)

Figure 7: Evaluation results of the baselines and CalibRAG utilizing two retriever models: BM25 (a, b) and Contriever (c, d) on NQ (a, c) and WebQA (b, d). Here, we utilize `DeepSeek-Distill` [59] as our user model $U$. We report four metrics—1-AUROC, 1-ACC, ECE, and Brier Score—where lower values indicate better performance.

Table 7: Comparison of fine-tuned RAG reranking methods using our synthetic training data on HotpotQA.

| Methods | AUROC ($\uparrow$) | ACC ($\uparrow$) | ECE ($\downarrow$) | BS ($\downarrow$) |
|---|---|---|---|---|
| Cross-encoder | 60.74 | 34.98 | 0.477 | 0.477 |
| Cross-encoder (Fine-tuned) | 61.55 | 32.54 | **0.008** | 2.555 |
| CalibRAG | **72.47** | **42.37** | 0.106 | **0.206** |

Table 8: Evaluation metrics of CalibRAG without reranking on WebQA using BM25

| Methods | AUROC ($\uparrow$) | ACC ($\uparrow$) | ECE ($\downarrow$) | BS ($\downarrow$) |
|---|---|---|---|---|
| Number | 69.38 ± 2.84 | 36.04 ± 0.50 | 0.1931 ± 0.0131 | 0.2293 ± 0.0102 |
| CalibRAG w/o Rerank | **75.73** ± 0.00 | **41.99** ± 0.03 | **0.0780** ± 0.0312 | **0.1981** ± 0.0025 |

## D.5   Ablation on Fine-Tuning for Reranking Baselines

To ensure a fair comparison between CalibRAG and the reranking baseline, we also evaluated a fine-tuned reranker model which fine-tuned using our synthetic datasets. However, as discussed in § 3.2, training was challenging due to the difficulty in feature extraction without using an embedding model to generate the guidance variable $z$. And this difficulty let fine-tuned model underfit to the training dataset. As shown in Table 7, the reranker model underperforms compared to the zero-shot setting. Therefore, in the **Comparison with reranking and robust RAG baselines** experiments in Sec. 4.1, we evaluated the CalibRAG model alongside zero-shot reranker models.

## D.6   Ablation on CalibRAG without reranking

To isolate the effect of reranking in our confidence calibration framework, we evaluated CalibRAG without using any reranking, where the model directly uses retrieved contexts without reordering them based on predicted confidence. As shown in Table 8, even without reranking, CalibRAG substantially outperforms the Number baseline in both accuracy and calibration metrics. These results

Table 9: Evaluation metrics of Number + Rerank and CalibRAG on WebQA

| Retriever | Methods | AUROC ($\uparrow$) | ACC ($\uparrow$) | ECE ($\downarrow$) | BS ($\downarrow$) |
|---|---|---|---|---|---|
| BM25 | Number + Rerank | 75.06 ± 0.00 | 42.42 ± 0.01 | 0.2075 ± 0.0167 | 0.2397 ± 0.0109 |
| | CalibRAG | **77.29** ± 0.42 | **43.77** ± 0.54 | **0.0567** ± 0.0332 | **0.1983** ± 0.0045 |
| Contriever | Number + Rerank | **76.84** ± 0.00 | 43.08 ± 0.00 | 0.2088 ± 0.0127 | 0.2390 ± 0.0083 |
| | CalibRAG | 76.24 ± 0.37 | **44.19** ± 2.60 | **0.0997** ± 0.0122 | **0.2095** ± 0.0062 |

Table 10: Comparison of zero-shot evaluation of calibration baselines on **NQ** and **WebQA** datasets using BM25 (lexical) retrieval. Results are averaged over three random seeds.

| Methods | NQ | | | | WebQA | | | |
|---|---|---|---|---|---|---|---|---|
| | AUROC | ACC | ECE | BS | AUROC | ACC | ECE | BS |
| *CT-LoRA* | 73.51 ± 1.65 | 37.70 ± 0.28 | 0.2479 ± 0.024 | 0.2709 ± 0.0133 | 74.36 ± 1.17 | 38.09 ± 0.28 | 0.2487 ± 0.0303 | 0.2681 ± 0.0200 |
| *CT-probe* | 60.92 ± 0.94 | 37.59 ± 3.03 | 0.3490 ± 0.0236 | 0.3536 ± 0.0223 | 58.52 ± 2.51 | 37.75 ± 4.39 | 0.3491 ± 0.0329 | 0.3539 ± 0.0332 |
| *Linguistic-LoRA* | 57.12 ± 4.35 | 39.42 ± 0.94 | 0.4529 ± 0.0223 | 0.4362 ± 0.0284 | 56.44 ± 1.93 | 40.58 ± 1.18 | 0.4536 ± 0.0071 | 0.4385 ± 0.0091 |
| *Number-LoRA* | 67.48 ± 1.42 | 34.38 ± 0.71 | 0.1922 ± 0.0165 | 0.2294 ± 0.0076 | 69.38 ± 2.84 | 36.04 ± 0.50 | 0.1931 ± 0.0131 | 0.2293 ± 0.0102 |
| *CalibRAG* | **77.29** ± 0.12 | 42.66 ± 0.97 | **0.0600** ± 0.0039 | **0.1983** ± 0.0017 | **77.29** ± 0.42 | 43.77 ± 0.54 | **0.0567** ± 0.0332 | **0.1983** ± 0.0045 |
| *CalibRAG-multi* | 76.73 ± 0.22 | **46.16** ± 0.05 | 0.1397 ± 0.0022 | 0.2138 ± 0.0016 | 76.40 ± 0.28 | **45.84** ± 0.25 | 0.1372 ± 0.0007 | 0.2175 ± 0.0008 |

Table 11: Comparison of zero-shot evaluation of calibration baselines on **NQ** and **WebQA** datasets using Contriever (dense) retrieval. Results are averaged over three random seeds.

| Methods | NQ | | | | WebQA | | | |
|---|---|---|---|---|---|---|---|---|
| | AUROC | ACC | ECE | BS | AUROC | ACC | ECE | BS |
| *CT-LoRA* | 69.89 ± 4.94 | 39.93 ± 1.26 | 0.2800 ± 0.0585 | 0.3008 ± 0.0435 | 69.81 ± 6.82 | 37.83 ± 1.25 | 0.2646 ± 0.0510 | 0.2860 ± 0.0394 |
| *CT-probe* | 63.84 ± 6.14 | 37.92 ± 2.80 | 0.3225 ± 0.0634 | 0.3343 ± 0.0498 | 62.65 ± 8.10 | 36.43 ± 4.03 | 0.3072 ± 0.0670 | 0.3180 ± 0.0565 |
| *Linguistic-LoRA* | 57.05 ± 3.91 | 41.50 ± 0.37 | 0.4368 ± 0.0267 | 0.4252 ± 0.0290 | 56.30 ± 2.70 | 39.76 ± 0.77 | 0.4657 ± 0.0124 | 0.4477 ± 0.0162 |
| *Number-LoRA* | 71.16 ± 0.61 | 35.99 ± 0.54 | 0.1827 ± 0.0124 | 0.2214 ± 0.0016 | 73.47 ± 1.01 | 35.61 ± 0.12 | 0.1754 ± 0.0124 | 0.2141 ± 0.0040 |
| *CalibRAG* | **73.89** ± 1.50 | 46.55 ± 2.45 | **0.0312** ± 0.0073 | **0.2074** ± 0.0062 | **76.24** ± 0.37 | 44.19 ± 2.60 | **0.0970** ± 0.0122 | **0.2095** ± 0.0062 |
| *CalibRAG-multi* | 72.73 ± 0.08 | **49.42** ± 0.07 | 0.1656 ± 0.0019 | 0.2375 ± 0.0013 | 72.95 ± 0.08 | **46.78** ± 0.02 | 0.1901 ± 0.0012 | 0.2488 ± 0.0009 |

indicate that the learned calibration itself, without requiring reranking, still provides significant benefit, demonstrating the robustness of CalibRAG's alignment mechanism.

### D.7  Using Uncertainty baseline confidence scores for reranking

In this section, we investigate the effectiveness of using uncertainty baseline confidence scores for reranking in the RAG pipeline. As described in the main paper, these confidence scores are derived from verbalized scalar predictions generated by the LLM, typically representing values from 0 to 100.

While such scalar confidence values can be used to rerank retrieved documents, this approach incurs significant computational overhead. Specifically, the Number baseline requires generating full guidance $z$ for every $(q, d)$ pair before estimating confidence, as the model conditions on both the query and document to generate scalar outputs. In contrast, CalibRAG directly estimates confidence from the $(q, d)$ pair using a lightweight forecasting function $f(q, d)$, thus avoiding this expensive intermediate generation.

Despite this additional cost, we performed an ablation to compare the reranking performance of Number-based confidence scores versus CalibRAG. As shown in Table 9, CalibRAG consistently outperforms the baseline across both BM25 and Contriever retrievers on the WebQA dataset.

These results demonstrate that CalibRAG not only provides better-calibrated decisions but does so more efficiently without requiring guidance generation for every document candidate. This highlights the dual advantage of CalibRAG in both performance and computational cost.

Table 12: Comparison of zero-shot evaluation of calibration baselines on **BioASQ-Y/N**, **MMLU-Med**, and **PubMedQA** datasets. Results are averaged over three random seeds.

| Methods | Dataset | AUROC | ACC | ECE | BS |
|---|---|---|---|---|---|
| *CT-LoRA* | BioASQ-Y/N | 65.20 $\pm$ 2.32 | 54.31 $\pm$ 0.73 | 0.5167 $\pm$ 0.012 | 0.5099 $\pm$ 0.0146 |
| | MMLU-Med | 66.94 $\pm$ 0.68 | 47.20 $\pm$ 1.52 | 0.4293 $\pm$ 0.0088 | 0.4262 $\pm$ 0.0084 |
| | PubMedQA | 56.67 $\pm$ 3.16 | 43.80 $\pm$ 0.91 | 0.4307 $\pm$ 0.0099 | 0.4300 $\pm$ 0.0094 |
| *CT-probe* | BioASQ-Y/N | 59.73 $\pm$ 4.27 | 57.98 $\pm$ 1.45 | 0.5664 $\pm$ 0.009 | 0.5630 $\pm$ 0.0094 |
| | MMLU-Med | 55.39 $\pm$ 3.24 | 49.49 $\pm$ 5.00 | 0.4771 $\pm$ 0.0384 | 0.4758 $\pm$ 0.0375 |
| | PubMedQA | 54.56 $\pm$ 0.61 | 46.60 $\pm$ 1.88 | 0.4506 $\pm$ 0.012 | 0.4510 $\pm$ 0.0121 |
| *Linguistic-LoRA* | BioASQ-Y/N | 48.24 $\pm$ 2.26 | 57.82 $\pm$ 0.50 | 0.3193 $\pm$ 0.0027 | 0.3464 $\pm$ 0.0030 |
| | MMLU-Med | 51.30 $\pm$ 0.93 | 55.43 $\pm$ 0.94 | 0.3262 $\pm$ 0.0078 | 0.3544 $\pm$ 0.0049 |
| | PubMedQA | 49.13 $\pm$ 0.79 | 47.13 $\pm$ 2.25 | 0.4047 $\pm$ 0.0336 | 0.4225 $\pm$ 0.021 |
| *Number-LoRA* | BioASQ-Y/N | 52.43 $\pm$ 2.19 | 53.72 $\pm$ 1.69 | 0.4664 $\pm$ 0.0355 | 0.4659 $\pm$ 0.0332 |
| | MMLU-Med | 53.47 $\pm$ 2.54 | 41.44 $\pm$ 1.01 | 0.3394 $\pm$ 0.0168 | 0.3541 $\pm$ 0.0135 |
| | PubMedQA | 50.34 $\pm$ 0.25 | 43.60 $\pm$ 0.59 | 0.3866 $\pm$ 0.0029 | 0.3954 $\pm$ 0.0032 |
| *CalibRAG* | BioASQ-Y/N | **66.66** $\pm$ 1.34 | **70.82** $\pm$ 3.34 | **0.2414** $\pm$ 0.0427 | **0.2606** $\pm$ 0.0386 |
| | MMLU-Med | **68.93** $\pm$ 1.32 | **57.20** $\pm$ 0.21 | **0.0625** $\pm$ 0.0653 | **0.2226** $\pm$ 0.0112 |
| | PubMedQA | **66.57** $\pm$ 2.00 | **62.20** $\pm$ 3.53 | **0.2250** $\pm$ 0.0353 | **0.2691** $\pm$ 0.0072 |

---

**Original question:** The American Sweetgum is the hostplant of what kind of bug?
**Open-ended query:** Write a paragraph about the kind of bug that uses the American Sweetgum as a host plant.

**Answer:** moth

**Original Retrieval Model's Top-1 document** (This context set the LM model's confidence to 'Confident'):
… American sweet gum, and other deciduous trees), and trapped in molasses-baited jars. One researcher who collected specimens extensively found it to be the most adaptable of the "Parcoblatta" species, trapping adults among logs and undergrowth on the borders of woodland areas, and taking specimens from pasture grasses, in grass under backyard trees, under dried cow dung, under trash and debris at woodland campsights, and from homes in wooded areas, which the species is sometimes reported to invade. Parcoblatta divisa Parcoblatta divisa, the southern wood cockroach, is a species of cockroach native to the United States.
 **User decision: The American Sweetgum is the host plant of the Parcoblatta divisa, also known as the southern wood cockroach.**

**CalibRAG's Top-1 document** (This context set the CalibRAG's confidence to 81.41):
… sometimes as imitation mahogany or Circassian walnut. It is used widely today in flake and strand boards. Sweetgum is a foodplant for various Lepidoptera caterpillars, such as the gypsy moth. The American sweetgum is widely planted as an ornamental, within its natural range and elsewhere. The hardened sap, or gum resin, excreted from the wounds of the sweetgum, for example, the American sweetgum ("Liquidambar styraciflua"), can be chewed on like chewing gum and has been long used for this purpose in the Southern United States. The sap was also believed to be a cure for sciatica, weakness of nerves, etc.
**User decision: The American Sweetgum is the host plant of the gypsy moth.**

Figure 8: Qualitative comparison of original retrieval model from CalibRAG.

## D.8 Full numerical results for main experiments

 present the complete numerical results from the primary experiments. For the *Base* model, we utilized a pretrained model, sampling sentences across three different seeds. For the other methods, training was conducted across three random seeds to ensure robust evaluation. We highlight the best-performing value in **bold** and the second-best in underline.

## D.9 Qualitative Results

While quantitative metrics alone may not fully capture all the benefits of CalibRAG, we present examples highlighting its ability to identify relevant documents and assign calibrated confidence scores. Given the query "Write a paragraph about the kind of bug that uses the American Sweetgum as a host plant.", the base retriever focuses only on the keyword "American Sweetgum,", retrieving

loosely relevant content and marking its confidence as 'Confident' (10/11) as illustrated in Fig. 8. This led to the incorrect conclusion that the sweetgum is the host plant of Parcoblatta divisa, the southern wood cockroach. In contrast, CalibRAG captures the full context, retrieving documents specifically about the gypsy moth, which uses the sweetgum as a host plant, and correctly assigns a confidence level of 81.41. This demonstrates the capability of CalibRAG to find a relevant document and assign a confidence level correlated with the accuracy of the downstream surrogate user. Additional examples can be found in § E.

## E  Data Examples

Fig. 9 shows the top 20 examples of queries and their corresponding labels. The full set of data examples will be released upon publication of the paper. Fig. 9 shows that the ranking of the retrieved documents is not correlated with the accuracy of the user decision. As seen in this example, the top-ranked document is not helpful for the user model in decision-making, whereas the second-ranked document provides information that can lead the user model to make a correct decision. This illustrates the importance of CalibRAG's forecasting function $f$ in effectively modeling the probability that a decision made using document $d$ is correct, emphasizing the need for reranking documents based on this modeling.

Open-ended query: Write a paragraph about the founding year of Apple Computer.

line became a sales smash, moving about one million units each year. It also helped re-introduce Apple to the media and public, and announced the company's new emphasis on the design and aesthetics of its products. In 1999, Apple introduced the Power Mac G4, which utilized the Motorola-made PowerPC 7400 containing a 128-bit instruction unit known as AltiVec, its flagship processor line. Also that year, Apple unveiled the iBook, its first consumer-oriented laptop that was also the first Macintosh to support the use of Wireless LAN via the optional AirPort card that was based on the 802.11b standard; it helped **(False)**

Xcode. Its online services include the iTunes Store, the iOS App Store and Mac App Store, Apple Music, and iCloud. Apple was founded by Steve Jobs, Steve Wozniak, and Ronald Wayne in April 1976 to develop and sell Wozniak's Apple I personal computer. It was incorporated as Apple Computer, Inc., in January 1977, and sales of its computers, including the Apple II, grew quickly. Within a few years, Jobs and Wozniak had hired a staff of computer designers and had a production line. Apple went public in 1980 to instant financial success. Over the next few years, Apple shipped new **(True)**

had told him about it and had said he needed the money, Wozniak would have given it to him. In 1975, Wozniak began designing and developing the computer that would eventually make him famous, the Apple I. On June 29 of that year, he tested his first working prototype, displaying a few letters and running sample programs. It was the first time in history that a character displayed on a TV screen was generated by a home computer. With the Apple I, he and Jobs were largely working to impress other members of the Palo Alto-based Homebrew Computer Club, a **(False)**

at the Homebrew Computer Club. Apple I was sold as a motherboard (with CPU, RAM, and basic textual-video chips), which was less than what is now considered a complete personal computer. The Apple I went on sale in July 1976 and was market-priced at $666.66 ($ in dollars, adjusted for inflation). Apple Computer, Inc. was incorporated on January 3, 1977, without Wayne, who left and sold his share of the company back to Jobs and Wozniak for $800 only a couple weeks after co-founding Apple. Multimillionaire Mike Markkula provided essential business expertise and funding of $250,000 during the incorporation of **(True)**

.
.
.

. Apple. During the first five years of operations revenues grew exponentially, doubling about every four months. Between September 1977 and September 1980, yearly sales grew from $775,000 to $118million, an average annual growth rate of 533%. The Apple II, also invented by Wozniak, was introduced on April 16, 1977, at the first West Coast Computer Faire. It differed from its major rivals, the TRS-80 and Commodore PET, because of its character cell-based color graphics and open architecture. While early Apple II models used ordinary cassette tapes as storage devices, they were superseded by the introduction of a -inch floppy **(True)**

of "Kilobaud Microcomputing", publisher Wayne Green stated that "the best consumer ads I've seen have been those by Apple. They are attention-getting, and they must be prompting sale." In August, the "Financial Times" reported that On December 12, 1980, Apple launched the Initial Public Offering of its stock to the investing public. When Apple went public, it generated more capital than any IPO since Ford Motor Company in 1956 and instantly created more millionaires (about 300) than any company in history. Several venture capitalists cashed out, reaping billions in long-term capital gains. In January 1981, Apple held its first shareholders . **(False)**

with no programming language built-in. This presented a problem to Apple: the Mac was due to be launched in 1983 (originally), with a new user interface paradigm, but no third-party software would be available for it, nor could users easily write their own. Users would end up with a computer that did nothing. In order to fill this void, several members of the Mac team took it upon themselves to write simple applications to fill these roles until third-party developers published more full-fledged software. The result was MacWrite and MacPaint, which shipped free with every Macintosh from 1984 to 1986. **(False)**

idea was to design technology based on a profile that included diskless computers, commonly coded applications using languages such as Java, and interface with the internet using common software such as Netscape Navigator. In May 1996, Apple became a partner in the network computing effort, and used the Apple Pippin platform as its implementation. On July 9, 1997, Gil Amelio was ousted as CEO of Apple by the board of directors. Steve Jobs stepped in as the interim CEO ("iCEO", as he was referred to), to begin a critical restructuring of the company's product line. He would eventually become CEO **(False)**

Apple Writer Apple Writer is a word processor for the Apple II family of personal computers. It was created by Paul Lutus and published in 1979 by Apple Computer. Paul Lutus wrote "Apple Writer" alone in a small cottage he built himself atop a hill in the woods of Oregon, connected to the electricity grid via of cable strung in trees. The original 1979 version of "Apple Writer" ran from a 13-sector DOS 3.2 diskette and supported 40-column text display. It displayed text entirely in uppercase, but case could be toggled by pressing the ESC key; characters that the user **(False)**

Figure 9: Top-20 retrieved document examples.

# F Qualitative Examples

Here, we present additional qualitative examples for comparison with other baselines. In Fig. 10, Fig. 11, Fig. 12, and Fig. 13, the examples demonstrate that while the baselines retrieve documents that provide incorrect answers to the queries, they still assign high confidence to the retrieved documents. In contrast, CalibRAG effectively reranks and retrieves documents that are highly relevant to the decision problem $x$, allowing us to confirm that the guidance generated from these retrieved documents is well-predicted to be helpful for decision-making. Additionally, we can confirm that when the document with the highest rank does not aid in decision-making for $x$, CalibRAG successfully assigns a lower confidence level, helping to prevent the user from over-relying on the guidance.

> **Original question:** When was the American lawyer, lobbyist and political consultant who was a senior member of the presidential campaign of Donald Trump born?
> **Open-ended query:** Write a paragraph about the American lawyer, lobbyist, and political consultant who was a senior member of Donald Trump's presidential campaign, including details about when he was born.
>
> **Answer:** April 1, 1949

> **Original Retrieval Model's Top-1 document** (This context set the LM model's confidence to 'Certain'):
> … Page testified that he did not "directly" express support for lifting the sanctions during the meeting with Baranov, but that he might have mentioned the proposed Rosneft transaction. Carter Page Carter William Page (born June 3, 1971) is an American petroleum industry consultant and a former foreign-policy adviser to Donald Trump during his 2016 Presidential election campaign.
> **User decision: June 3, 1971.**

> **CalibRAG's Top-1 document** (This context set the CalibRAG's confidence to 83.93):
> Paul Manafort Paul John Manafort Jr. (born April 1, 1949) is an American lobbyist, political consultant, lawyer, and felon. A Republican, he joined Donald Trump's presidential campaign team in March 2016, and was campaign chairman from June to August 2016. Manafort was an adviser to the U.S. presidential campaigns of Republicans Gerald Ford, Ronald Reagan, George H. W. Bush, and Bob Dole. In 1980, he co-founded the Washington, D.C.-based lobbying firm Black, Manafort & Stone, along with principals Charles R. Black Jr., and Roger J. Stone, joined by Peter G. Kelly in 1984. Manafort often lobbied on behalf of …
> **User decision: April 1, 1949.**

Figure 10: **CalibRAG vs Linguistic-LoRA.** In the case of CalibRAG, a document about the person in question is retrieved with a confidence level of 83.93%. In contrast, the document retrieved by the base retrieval model is related to Donald Trump but does not match the specific person in the query. Nevertheless, the Linguistic-LoRA model trust the document confidently.

> **Original question:**
> Where did the allies go after north Africa?
> **Open-ended query:**
> Write a paragraph about the next steps taken by the Allies following their involvement in North Africa.
>
> **Answer:** Italy

> **Original Retrieval Model's Top-1 document** (This context set the LM model's confidence to 6/10):
> … Abteilung 350 procedures, subsequently the German High Command was completely taken by surprise by the bulk of the Allies Mediterranean theatre buildup and subsequent landing in French North Africa on 7 November 1942, that was known by the allies as Operation Torch. Not until the Tunisian Campaign was well under way did W-Leit 2 at Taormina pull itself together and contribute to the general effort by intercepting air support messages. It is worth noting that General Albert Praun, who was chief Signals of the Wehrmacht thought that the Allied Invasion of French North Africa was not detected due to radio…
> **User decision: Based on the context provided, the Allies advanced towards Tunisia after North Africa.**

> **CalibRAG's Top-1 document** (This context set the CalibRAG's confidence to 77.98):
> 1943, at the Casablanca Conference, the Allies agreed to demand Axis unconditional surrender, a war aim which implied the physical occupation of Germany with land forces. While building up strength for an invasion of continental Europe, the Allies pursued an indirect strategy by invading Europe from the South. After defeating Axis forces in North Africa (the invasion of French North-Africa), Sicily and southern Italy were invaded, leading to the defeat of Fascist Italy. Churchill especially favoured a Southern strategy, aiming to attack the "soft underbelly" of Axis Europe through Italy, Greece and the Balkans in a strategy similar to the> Question: Write a paragraph about the next steps taken by the Allies following their involvement in North Africa.
> **User decision: The Allies went to Sicily.**

Figure 11: **CalibRAG vs Number-LoRA.** In the case of CalibRAG, an accurate document about the location following North Africa was retrieved, allowing the user model to make a correct decision. In contrast, the base retrieval model brought a different document. Nevertheless, Number-LoRA model set this context with a confidence level of 6 out of 10, leading the user to ultimately make an incorrect decision.

**Original question:**
Who is the first president to be impeached?
**Open-ended query:**
Write a paragraph about the first president to be impeached.

**Answer:** Andrew Johnson

**Original Retrieval Model's Top-1 document** (This context set the LM model's confidence to 9/10):
Constitution and the whole history of our republic demand that we make up our minds." The Judiciary Committee, with six Republicans joining the Democratic majority, passed three of the five proposed articles of impeachment. On July 27, 1974, the Committee voted 27–11 to recommend the first article of impeachment against the president: obstruction of justice. The Committee then recommended the second article, abuse of power, on July 29, 1974. The next day, on July 30, 1974, the Committee recommended the third article: contempt of Congress. Article I alleged in part: On June 17, 1972, and prior thereto, agents of the...
**User decision: Richard Nixon.**

**CalibRAG's Top-1 document** (This context set the CalibRAG's confidence to 92.32):
... justice following allegations that he committed perjury and obstructed justice to conceal an affair he had with Monica Lewinsky, a 22-year old White House intern. Clinton was acquitted by the Senate in 1999 and completed his term in office. He is only the second U.S. president to ever be impeached, the first being Andrew Johnson. During the last three years of Clinton's presidency, the Congressional Budget Office reported a budget surplus, the first such surplus since 1969. In foreign policy, Clinton ordered U.S. military intervention in the Bosnian and Kosovo wars, signed the Iraq Liberation Act in opposition to Saddam ...
**User decision: Andrew Johnson.**

Figure 12: **CalibRAG vs Number-LoRA.** The base retrieval model focused solely on the word 'impeached' and retrieved a related document, missing the context of 'first.' Despite this, the Number-LoRA model set a confidence level of 9 out of 10, causing the user to make an incorrect decision. In contrast, CalibRAG retrieved an accurate document that, while not explicitly containing 'first impeached,' included the phrase 'first being.' It set a confidence level of 92.32%, allowing the user to arrive at the correct answer.

**Original question:**
what year is it for the Jewish calendar?
**Open-ended query:**
Write a paragraph about the current year in the Jewish calendar.

**Answer:** AM 5778

**Original Retrieval Model's Top-1 document** (This context set the LM model's confidence to 96.83):
...of two or three Hebrew letters. See Four gates. In Hebrew there are two common ways of writing the year number: with the thousands, called ("major era"), and without the thousands, called ("minor era"). Thus, the current year is written as ' () using the "major era" and ' (%1000) using the "minor era". In 1178 CE, Maimonides wrote in the "Mishneh Torah", "Sanctification of the Moon" (11.16), that he had chosen the epoch from which calculations of all dates should be as "the third day of Nisan in this present year ... which is the year 4938 of the...
**User decision: According to the provided context, the current year in the Hebrew calendar is written as 5784 (with the thousands) and 2444 (without the thousands). Given the model's high confidence level.**

**CalibRAG's Top-1 document** (This context set the CalibRAG's confidence to 20.95):
...year; and about every 231 years it will fall a day behind the mean Gregorian calendar year. The era used since the Middle Ages is the epoch (Latin for "in the year of the world"; , "from the creation of the world"). As with (A.D. or AD), the words or abbreviation for (A.M. or AM) for the era should properly "precede" the date rather than follow it. AM began at sunset on and will end at sunset on . The Jewish day is of no fixed length. The Jewish day is modeled on the reference to "...there was evening and...
**User decision: I'm not able to provide the current year in the Jewish calendar based on the given context and the model's confidence level. The model's confidence level is 20.95, which is below.**

Figure 13: **CalibRAG vs CT-LoRA.** In the case of CalibRAG, the top-20 confidence score is 20.95 for incorrect information, causing the user to hesitate in making a decision. However, with the CT-LoRA model, incorrect information is assigned a confidence score of 96.83, leading the user to make an incorrect decision.

# G Prompt Examples

In this section, we present prompt examples used during training and inference. Figure 14a shows the prompt that encourages the user model $U$ to act like a human decision-maker, leading it to over-rely on the guidance provided by the LLM. Figure 14b displays the prompt that generates the open-ended query $q$ from the decision task $x$. Figure 14c presents the prompt that induces the generation of guidance $z$ from $M$ based on the retrieved document $d$. Figure 15a is used when grading the user model $U$'s decision against the true answer using $\mathcal{G}$. Figure 16a, Figure 16b, and Figure 16c are prompts used to instruct $\mathcal{M}$ to generate confidence in terms of linguistic or numerical calibration. Lastly, Figure 15b is the prompt used during **Stage 3** of the inference process.

---

**Decision prompt**

```
The task is to answer questions based on a context generated
by a language model in response to a question about relevant
information, along with the model's confidence level in the
provided answer.
Context:   {context}
Question:   {question}
Model Confidence:   {confidence}
Answer:
```

(a) Prompt designed to guide the user model $U$ in making decisions based on the LLM-generated guidance $z$ and confidence $c$.

---

**Prompt that generates open-ended query $q$ from the decision task $x$**

```
You are an automated assistant tasked with rephrasing specific
questions into open-ended queries to encourage detailed exploration
and discussion of the key topics mentioned.
Your goal is to prompt someone to write a paragraph exploring the
topic without directly revealing the answer.
Examples for Guidance:
Example 1:
Question 1:  Which sea creature is the world's largest
invertebrate?
Question 2:  Write a paragraph about the world's largest
invertebrate.
...
Now, please rephrase the following question:
Question 1:  {question}
Question 2:
```

(b) This prompt was first suggested by Band et al. [6], and we have modified part of the proposed prompt for our use here. We use this prompt as an input when generating the query $q$ based on the decision task $x$.

---

**Guidance $z$ generation prompt**

```
Directly state the answer without phrases like 'the correct answer
is.
Given the retrieved context, answer the question as accurately as
possible.
Question:   {question}
Retrieved Context:   {title} - {context}
Answer:
```

(c) This prompt guides the LLM $\mathcal{M}$ to provide direct, concise guidance $z$ based on a given retrieved document $d$.

Figure 14: Prompt used for (a) user model making decisions, (b) generating $q$ from $x$, and (c) generating $z$.

**Evaluation prompt**

```
The problem is:  {question} The correct answer for this problem is:
{ground-truth}  A student submitted the answer: {prediction}  The
student's answer must be correct and specific but not overcomplete
(for example, if they provide two different answers, they did not
get the question right).
However, small differences in formatting should not be penalized
(for example, 'New York City' is equivalent to 'NYC').     Did the
student provide an equivalent answer to the ground truth?  Please
answer yes or no without any explanation:
```

(a) Prompt used to evaluate the long-form generated answer.

**Query regeneration prompt.**

```
You are a language model assistant who specializes in improving
queries for document search systems.  Your task is to highlight
and clarify the important parts of a given query to make it more
precise and help retrieve relevant documents.
Please take the original search query below and rewrite it by
emphasizing the important words.  Do not add any new information
not included in the original query.
Original Retrieval Query:  {query}
Please generate the new retrieval query without any explanation:
```

(b) This prompt assists in rewriting search queries to enhance precision and relevance for document retrieval, emphasizing the crucial elements without adding extraneous information.

Figure 15: Prompt used for (a) evaluation and (b) query regeneration.

**Calibration tuning prompt**

```
Is the proposed answer correct?
Choices:
(i):  no
(ii): yes
Answer:
```

(a) This prompt was first suggested by Kapoor et al. [21]. It poses a straightforward question to verify the correctness of a proposed answer with binary choices for evaluation. We used this prompt when training our baselines.

**Linguistic calibration prompt**

```
Provide the certainty level of answer using the given 11 certainty
levels.  Give ONLY your certainty level, no other words or
explanation.
Certainty Levels:  Unlikely, Doubtful, Uncertain, Ambiguous,
Probable, Likely, Possible, Specified, Confirmed, Certain,
Inevitable.
For example:  Certainty:  <ONLY the certainty level that Answer
is correct, without any extra commentary whatsoever; just the
certainty level!>
Certainty:
```

(b) This prompt requires the model to evaluate the certainty of an answer using a predefined set of linguistic levels of certainty. We used this prompt for our baselines that utilize linguistic calibration.

**Number calibration prompt**

```
Provide the certainty level of answer using the given 11 certainty
levels.  Give ONLY your certainty level, no other words or
explanation.
Certainty Levels:  0, 1, 2, 3, 4, 5, 6, 7, 8, 9, 10.
For example:  Certainty:  <ONLY the certainty level that Answer is
correct, without any extra commentary whatsoever; just the number!>
Certainty:
```

(c) This prompt is similar to the linguistic calibration prompt but uses numerical certainty levels (from 0 to 10) to rate the confidence in the answer provided. We used this prompt for our baselines that utilize number calibration.

Figure 16: Prompt used for baseline experiments.

