# OpenReview forum: "Reliable Decision‑Making via Calibration‑Oriented Retrieval‑Augmented Generation"
_NeurIPS.cc/2025/Conference — NeurIPS 2025 poster_

### Official Review · Reviewer_xGBU · 2025-06-16

**Clarity:** 3
**Significance:** 3
**Originality:** 3
**Rating:** 5
**Confidence:** 4

**Summary:**

This paper introduces CalibRAG, a Retrieval-Augmented Generation (RAG) framework designed to produce not only relevant evidence for decision-making but also well-calibrated confidence estimates for those decisions. Traditional RAG approaches often yield overconfident guidance, whereas CalibRAG trains a forecasting function that, given a prompt, a retrieved document, and a model temperature setting, predicts the probability of a correct decision. Empirical results across several question-answering tasks and retrieval back ends demonstrate that CalibRAG improves decision accuracy and calibration metrics relative to standard baselines.

**Questions:**

1. Do you foresee applying CalibRAG to other generative tasks such as summarization or code generation?
2. Retrieving and reranking a sizable set of documents and performing query reformulation adds computational cost. Have you analyzed the detailed inference time overhead?

**Ethical Concerns:**

["NO or VERY MINOR ethics concerns only"]

**Final Justification:**

I have read the rebuttal, which has fully resolved my concerns. I decide to keep my score positive.

**Limitations:**

See weakness above

**Quality:**

4

**Strengths And Weaknesses:**

Strengths:
1. The paper clearly articulates the issue of overconfident retrieval in RAG systems, using intuitive examples that ground the reader in the practical need for decision calibration.
2. Integrating a temperature-aware forecasting function into the retrieval step is novel and makes sense.
3. The evaluation spans varied question-answering benchmarks and diverse retrieval back ends.

Weakness:
1. Retrieving and reranking a sizable set of documents, with potential query reformulation, may introduce non-trivial latency.
2. The forecasting function is trained on synthetic data derived from a surrogate language model, leaving open questions about how well it mirrors real human decision behavior.

---

> ### Author Rebuttal · Authors · 2025-07-31
>
> >**[W1, Q2] Retrieving and reranking a sizable set of documents, with potential query reformulation, may introduce non-trivial latency. Retrieving and reranking a sizable set of documents and performing query reformulation adds computational cost. Have you analyzed the detailed inference time overhead?**
>
> Recent LLM-based reranking methods [1,2,3] typically require feeding multiple documents into the LLM simultaneously to assess their relative importance. This involves generating token-level outputs over all N documents, making the comparisons relative and thus difficult to parallelize. Moreover, such approaches are sensitive to the context length limitations of LLMs. As a result, their time complexity is approximately O($N^2⋅L^2+NL^2$), where L is the average document length, and if we assume that the length of the generated token is also L.
>
> In contrast, CalibRAG independently processes each query-document pair ($q$, $d_i$) to predict a confidence score. Here, the theoretical time complexity for the CalibRAG is O($N⋅L^2$), which is smaller than the recent LLM-based reranking methods. Furthermore, the confidence score is computed via a lightweight MLP without requiring autoregressive decoding, making it substantially faster and more efficient.
>
> To support this, we provide a direct comparison of inference time between CalibRAG and LLM reranking using the same document set in Table R.1, with interpretation to be included in the final version.
> Meanwhile, query reformulation is an optional component. For the sake of fair and consistent comparisons, we did not apply it in most experiments. It is only used once in Figure 5-(c) to explicitly assess its effect. In practical deployment, users may choose to apply reformulation depending on their desired trade-off between performance and computational cost.
>
> In summary, CalibRAG is designed for efficient document reranking and confidence estimation, and demonstrates clear advantages in terms of inference efficiency.
>
> __Table R.1__ Inference time comparison between CalibRAG and LLM-based reranking. We used Llama-3.1-8B-Instruct and measured the average inference time over 20 randomly selected queries from the NQ dataset. CalibRAG evaluates each (query, document) pair independently and in parallel, while LLM-rerank jointly processes all documents via ~500-token generation. Measurements were taken on a single A6000 48GB GPU.
>
> | Method | Time per document set
> -|-
> LLM-rerank | ~10 minutes |
> CalibRAG (avg) |~1.7 minutes |
>
> [1] Zhuang, Shengyao, et al. "A setwise approach for effective and highly efficient zero-shot ranking with large language models." Proceedings of the 47th International ACM SIGIR Conference on Research and Development in Information Retrieval. 2024.
>
> [2] Qin, Zhen, et al. "Large Language Models are Effective Text Rankers with Pairwise Ranking Prompting." Findings of the Association for Computational Linguistics: NAACL 2024. 2024.
>
> [3] Kim, Jaehyung, et al. "SuRe: Summarizing Retrievals using Answer Candidates for Open-domain QA of LLMs." The Twelfth International Conference on Learning Representations.
>
> >**[W2] The forecasting function is trained on synthetic data derived from a surrogate language model, leaving open questions about how well it mirrors real human decision behavior.**
>
> Thank you for the thoughtful comment. While it is true that the forecasting function is trained on synthetic data generated from a surrogate language model, we would like to highlight several key points that support the practicality and reliability of our approach.
>
> First, we conducted a human evaluation to assess how well the model’s predictions align with actual human decisions. Specifically, we randomly sampled 2 questions per confidence bin across 5 bins from WebQA, NQ, and BioASQ (totaling 10 examples), and presented the question and retrieved context to 10 human annotators. They were asked whether they agreed with the user model’s judgment—that is, whether the retrieved context was helpful for answering the question. The results showed an agreement rate of 81.3%, indicating that the surrogate-based forecasts meaningfully reflect human judgment.
>
> Second, conducting full-scale user studies where humans serve as decision-makers in downstream tasks is logistically challenging and costly. As an alternative, we simulate diverse user preferences by evaluating CalibRAG under multiple distinct surrogate user models (e.g., Phi-4, DeepSeek-Distill), which reflect different decision criteria and levels of background knowledge. As shown in Appendix D.4, CalibRAG consistently outperforms baselines across these varied simulated user types, demonstrating robustness to user variability.
>
> Third, although our current setup relies on LLM-based simulations, it can be naturally extended to incorporate real user data. For example, future work could fine-tune the forecasting model using expert annotations or crowd-sourced feedback to enhance realism and generalization.
>
> >**[Q1] Do you foresee applying CalibRAG to other generative tasks such as summarization or code generation?**
>
> Thank you for the question. We believe CalibRAG can be extended to other generative tasks such as summarization or code generation, as long as a meaningful retriever can be constructed. For summarization, this applies in settings where specific summary formats are desired and can be retrieved. For code generation, the applicability depends more on retrieving relevant algorithmic patterns, which is feasible but more domain-specific. Importantly, although our model was trained on general-domain data, it already shows strong generalization to out-of-domain tasks (e.g., medical QA), suggesting broader applicability of our framework.

---

> ### Comment · Reviewer_xGBU · 2025-08-06
>
> Thanks for your feedback! The rebuttal has fully resolved my questions and I decide to keep my score positive.

---

### Official Review · Reviewer_urW9 · 2025-06-20

**Clarity:** 3
**Significance:** 3
**Originality:** 2
**Rating:** 4
**Confidence:** 2

**Summary:**

The paper proposes to add calibration estimation to RAG outputs. CalibRAG incorporates a forecasting function—a surrogate model—that predicts the probability that a user's decision, informed by the retrieved information, will be accurate. They generate synthetic training data to teach the forecasting model to align confidence with actual decision outcomes and integrate it into the retrieval and response pipeline.

**Questions:**

na

**Ethical Concerns:**

["NO or VERY MINOR ethics concerns only"]

**Final Justification:**

It is borderline mainly because it sounds ok but not especially exciting or unusual and my low confidence in the field. But, it seems al reviewers lean towards accepting it, so can be accept as far as I am concerned.

**Quality:**

3

**Strengths And Weaknesses:**

Strengths:
The ideas is simple and introduces a clear extension borrowing from far enough field to this field.
The writing is clear, and the presentation is appealing.
The papers' scores seem high and the performance convincing.
The paper uses multiple baselines and retrievers to give a more round evaluation and validate the generality of their method.

Weakness:
The idea sounds very clean, so much so that it is surprising no one did it, despite so much RAG works. A search on my side didn't find something like that though.
I see the dataset as quite a minor contribution, but it is not a negative to have *also* a minor addition.

Minor:
The people in the figure are pretty, but maybe keeping mostly they faces would make the differences between them more salient?
In the contributions paragraph, references to where each part is written could be useful for people to jump to the right place (e.g., the dataset is not reflected in the section structure).

---

> ### Author Rebuttal · Authors · 2025-07-31
>
> >**[W1] Is the idea truly novel, given its simplicity, and is the dataset contribution substantial enough to strengthen the paper's overall impact?**
>
> Thank you for the kind feedback. We also found it surprising that prior RAG research has largely overlooked decision-aware confidence calibration, especially given the importance of reliability in downstream applications. We believe this is due to the lack of clear formulation and evaluation settings for decision-centric RAG scenarios, which our work aims to establish. We appreciate your recognition that our dataset, while a smaller contribution, serves as a concrete step toward enabling standardized evaluation in this space.
>
> >**[W2] The people in the figure are pretty, but maybe keeping mostly their faces would make the differences between them more salient? In the contributions paragraph, references to where each part is written could be useful for people to jump to the right place (e.g., the dataset is not reflected in the section structure).**
>
> Thank you for the helpful suggestions. We will revise the figure to focus more clearly on the distinctive facial features, which better convey diversity among user types. Additionally, we will edit the contributions paragraph to include references to the relevant sections for each component, including the dataset, to improve navigability and clarity in the final version.

---

> ### Comment · Reviewer_urW9 · 2025-08-01
>
> Great, note that while I gave suggestions, I will not be offended or anything if you choose some of them to be not helpful. (Of course I believe they are :-) )
>
> My score partially reflected my low confidence and the (hard to calibrate) impact, so I am leaning towards acceptance still, as far as I am concerned.

---

> > ### Author Response · Authors · 2025-08-03
> >
> > Thank you very much for your thoughtful feedback and constructive suggestions. We truly appreciate your engagement and are grateful that you are leaning toward acceptance. If the paper is accepted, we will carefully reflect and incorporate all of your suggestions into the final version.

---

### Official Review · Reviewer_gigS · 2025-06-21

**Clarity:** 2
**Significance:** 2
**Originality:** 3
**Rating:** 4
**Confidence:** 5

**Summary:**

This paper proposes CalibRAG, a method to align the confidence level of Retrieval-Augmented Generation (RAG) systems with actual correctness.  It first employs an LLM-based surrogate user model to create probability labels about what proportion of answers (sampled outputs conditioned on a temperature) with the retrieved documents are correct. Then, it trains a parameterized forecasting model that predicts the probability. In the inference stage, CalibRAG utilizes the forecasting model to rerank the retrieved documents and select the highest-ranked document for final generation. CalibRAG improves both accuracy and calibration performance over baselines in various QA datasets.

**Questions:**

## Questions

1. Is it possible to show the performance of CalibRAG on other types of "decision-making" tasks or real-user experiments? I would say **the limited evaluation (on QA tasks) is my main concern** since one can argue that in QA tasks, a RAG model can give answers as well, so it is not a necessity to have a "user model".
2. In the inference stage, you only select the top-1 document for the final generation. However, current RAG models (w/ long-context ability) can **process much more than 1 document, and it may naturally utilize the most useful information**, making the reranking less necessary. Can you provide some results about using more retrieved documents?
3. L263: For baselines, guidance and confidence are generated based on the top-1 document retrieved. Is it fairer to generate the confidence scores for all $K$ documents and utilize the most confident one?
4. As you use GPT-4o-mini as the evaluator in both your forecasting model training and final judgment, the forecasting model can somehow overfit (not sure, just guessing). Could you use another evaluator in the final judgment to show the robustness of CalibRAG if possible?
5. What is the reason for not taking guidance $z$ as one of the input params of the forecasting model? Is it because of efficiency? As your ultimate goal is to predict **how $z$ impacts decisions**, using $q$ and $d$ as substitutes seems suboptimal, as $z$ also incorporates the inherent knowledge of RAG models.
6. Why do you use Llama-2-7B in Table 1b, not aligned with the main setup (Llama-3.1-8B)?
7. L342: "gains diminish beyond 40 documents". However, it seems that using 40 documents yields larger gains compared to 20. Can you clarify this statement?
8. What model is used for the query reformulation? What are the maximum reformulation times when the threshold cannot be satisfied?

## Suggestions

Typos in writing:

1. L224: $t$.
2. L250: Change "decision model" to "user model".
3. Calibrag (should be CalibRAG) in Figure 5c.

[1] may be a missing related work (or baseline).

[1] Zhao, Xinran, et al. "Fact-and-Reflection (FaR) Improves Confidence Calibration of Large Language Models." Findings of ACL 2024.

**Ethical Concerns:**

["NO or VERY MINOR ethics concerns only"]

**Final Justification:**

The authors have largely addressed my concerns regarding the QA evaluation by providing a human study and presenting results using more RAG documents, where CalibRAG still demonstrates advantages. Overall, the method is reasonable and novel, with potential value to the field.

**Limitations:**

Yes.

**Paper Formatting Concerns:**

No.

**Quality:**

3

**Strengths And Weaknesses:**

## Strengths

1. The overall method design is reasonable. Using decoding temperature to reflect user behavioral traits is interesting.
2. The baselines included in the comparison are representative and comprehensive.

## Weaknesses

1. The evaluation in this paper is mainly about QA tasks, while decision-making (highlighted by this paper) may not only include QA.
2. All experiments consider using the top-1 document in the final generation, which seems a little bit far from the mainstream RAG pipeline.

---

> ### Author Rebuttal · Authors · 2025-07-31
>
> >**[W1, Q1]  Can CalibRAG demonstrate effectiveness on decision-making tasks beyond QA or in real-user scenarios?**
>
> We agree that QA tasks alone may not sufficiently capture the broader notion of decision-making, as direct answering can reduce the role of a user model. To address this, we designed our setup to simulate decision-making: instead of answering the original question $x$, CalibRAG generates an open-ended query $q$ to retrieve information relevant to user decisions (L195–L196). This aligns with recent work on LLM-based uncertainty calibration and decision support [1, 2].
>
> We acknowledge that surrogate user models cannot fully replicate real human behavior. To mitigate this, we employed diverse LLMs including LLaMA 2 7B (Table 1-(b)), Phi 4, and DeepSeek Distill (Appendix D.4) to reflect varying user backgrounds and preferences.
>
> Additionally, we conducted a human evaluation to assess how closely CalibRAG's judgments align with human decisions. Specifically, we sampled 2 questions per confidence bin across 5 bins from WebQA, NQ, and BioASQ (10 examples total), and asked 10 human raters whether they agreed with the user model’s judgment (i.e., whether the retrieved document was helpful). The result was an average agreement rate of 81.33%, suggesting strong alignment between surrogate model outputs and human intuition.
>
> Together, these quantitative and qualitative results demonstrate that CalibRAG goes beyond standard QA and effectively supports realistic decision-making scenarios.
>
> [1] Band, Neil et al. "Linguistic Calibration of Long-Form Generations." ICML, 2024.
>
> [2] Kapoor, Sanyam et al. "Large Language Models Must Be Taught to Know What They Don’t Know." NeurIPS, 2024.
>
> >**[W2, Q2] Recent RAG models can handle multiple documents at once. Why did you use only the top-1 document for generation? Wouldn’t using more documents reduce the need for reranking? Do you have any experiments evaluating this?**
>
> Thank you for the insightful question. In response to your suggestion, we conducted additional experiments to compare the performance of CalibRAG and the baseline (Number) in a multi-document RAG setting.
>
> Specifically, we used the BM25 on the WebQA, with Llama-3.1-8B-Instruct as the user model. In this setup, we provided the LLM with the top-k documents selected either by the retriever (baseline) or by CalibRAG’s confidence-based reranking, and generated final answers based on the combined input. As shown in Table R.1, even when using k=3 documents, CalibRAG consistently outperforms the baseline in terms of ACC as well as calibration metrics.
>
> Due to resource constraints, we limited our evaluation to k ≤ 3. Within this range, we observed that increasing k improves ACC for both methods. However, prior work [1] has shown that in multi-document RAG settings, the order of documents significantly affects performance, and [2] further reports that providing more than three documents may introduce contradictory information, which leads to increased noise and degraded performance. These findings suggest that simply adding more documents does not necessarily lead to better outcomes. **Instead, effective document selection and reranking remain essential.**
>
>
> __Table R.1__ Comparison of performance in multi-document generation settings.
>
> | 		           |  AUROC  |  ACC  |  ECE  | BS |
> |-|-|-|-|-|
> | Number   (k=1)         |   69.38    |   36.04    |    0.1931   |   0.2293   |
> | Number   (k=2)         |   66.46   |    38.90   |   0.2580    |   0.2888   |
> | Number   (k=3)         |   64.61    |    41.28   |   0.2907    |   0.3204   |
> | CalibRAG   (k=1)      |    77.29   |   43.77    |   0.0567    |    0.1983   |
> | CalibRAG   (k=2)      |   75.47    |   42.86    |   0.0557    |   0.2029    |
> | CalibRAG   (k=3)      |    76.75   |   43.28    |   0.0655    |   0.1999    |
>
> [1] Liu, Nelson F., et al. "Lost in the Middle: How Language Models Use Long Contexts." TACL., 2024.
>
> [2] Kim, Jaehyung, et al. "SuRe: Summarizing Retrievals using Answer Candidates for Open-domain QA of LLMs." ICLR., 2024.
>
>
> >**[Q3] L263: For baselines, guidance, and confidence are generated based on the top-1 document retrieved. Is it fairer to generate the confidence scores for all documents and utilize the most confident one?**
>
> As noted in L299–L301, we would like to emphasize that **we have already conducted this experiment.** As described in Appendix D.7, we explored an extended baseline where decoder outputs $z$ are generated for all retrieved documents. A separate confidence model is then used to predict confidence scores for each document, and the one with the highest score is selected. While this approach is conceptually valid, it suffers from significant computational overhead due to the need to generate responses for all documents. Despite this additional cost, the calibration performance of this baseline remained inferior to that of CalibRAG.
>
> __Table 9.__  Evaluation metrics of Number + Rerank and CalibRAG on WebQA
> | Retriever | Methods         | AUROC        | ACC         | ECE        | BS         |
> |-|-|-|-|-|-|
> | BM25      | Number + Rerank     | 75.06 | 42.42| 0.2075 | 0.2397 |
> |               | CalibRAG            | 77.29  | 43.77  | 0.0567| 0.1983   |
> | Contriever| Number + Rerank     | 76.84      | 43.08    | 0.2088   | 0.2390   |
> |               | CalibRAG            | 76.24  | 44.19  | 0.0997 | 0.2095 |
>
> *For completeness, we summarize the results from Appendix D.7-Table 9 again here.*
>
> >**[Q4] Could you use another evaluator in the final judgment to show the robustness of CalibRAG if possible?**
>
> Thank you for the suggestion. Table R.2 reports results on the NQ using BM25, and the evaluator used for decision scoring is LLaMA-3.2-3B-Instruct. We acknowledge the value of evaluating robustness across different evaluators and plan to include results on additional tasks with alternative models in the final version.
> Nonetheless, the clear and consistent improvements observed in all metrics indicate that CalibRAG exhibits robust calibration performance under the current setting.
>
> __Table R.2__ Evaluation on NQ dataset using BM25 and LLaMA-3.2-3B-Instruct as the evaluator.
> | Setting   | AUROC   | ACC     | ECE     | BS |
> |-|-|-|-|-|
> | CT        | 0.5201  | 0.4406  | 0.3104  | 0.3510       |
> | CT-Probe  | 0.6217  | 0.3963  | 0.3188  | 0.3426       |
> | Linguistic| 0.5964  | 0.4453  | 0.4262  | 0.4109       |
> | Number    | 0.6484  | 0.4312  | 0.3105  | 0.3367       |
> | Calibrag  | 0.7436  | 0.5196  | 0.0963  | 0.2204       |
>
>
>
> >**[Q5] What is the reason for not taking guidance  as one of the input params of the forecasting model?**
>
> As you noted, our goal is to predict how the generated guidance $z$ affects user decisions. While explicitly generating $z$ for every $(q, d)$ pair is computationally expensive, our model addresses this by using the internal representations of the LLM that generates $z$ as features (L160–161, L168), allowing it to capture $z$'s influence implicitly. This design preserves efficiency, and its effectiveness is validated by strong empirical performance.
>
> >**[Q6] Why do you use Llama-2-7B in Table 1b, not aligned with the main setup?**
>
> We used LLaMA-2-7B in Table-1-(b) for two main reasons.
> First, the Self-RAG model is only publicly available with LLaMA-2-7B and 13B, so we used the same base model to ensure a fair and reproducible comparison.
> Second, we aimed to show that CalibRAG performs robustly across different backbone architectures. The results in Table-1-(b) confirm that CalibRAG remains effective even when used with a different LLM.
>
> >**[Q7] L342: "gains diminish beyond 40 documents". However, it seems that using 40 documents yields larger gains compared to 20. Can you clarify this statement?**
>
> Thank you for the observation. We have conducted additional experiments on both WebQA and NQ using BM25 and Contriever as retrievers. Across the majority of these settings, we observed that performance gains tend to saturate beyond the top-20 retrieved documents. The statement in L342 was based on this general trend. However, Figure 5-(b) happens to show one case (WebQA with Contriever) where the performance improvement from 20 to 40 documents appears more pronounced, which may have caused the perceived inconsistency. We will revise the text and figure to clarify that gains generally diminish after 20 documents across most settings and support this with a more complete set of results in the final version.
>
> __Table R.3__ Performance of CalibRAG across different numbers of retrieved documents.
> | Setting / # Documents |   5   |  10   |  20   |  40   |
> |-|-|-|-|-|
> | BM25 - NQ             |  42.96     |   42.22   |   43.77    |   43.68    |
> | BM25 - WebQA          |   42.21    |    42.40   |   42.66    |   42.70    |
> | Contriever - NQ       |    44.27   |   45.05    |   46.55    |   46.07    |
>
> >**[Q8] What model is used for the query reformulation? What are the maximum reformulation times when the threshold cannot be satisfied?**
>
> Thank you for the question. The model used for query reformulation is the same LLM used to generate the guidance $z$. We perform query reformulation only once in the experiment shown in Table 2-(c), where we explicitly evaluate its effect. In principle, query reformulation can be applied repeatedly based on user configuration or an adaptive stopping criterion. However, in our main experiments, we did not apply reformulation at all in order to ensure fair and consistent comparison across all methods. We will clarify this implementation detail in the final version.
>
> **Additionally, thank you for the helpful suggestions. We will incorporate Zhao et al. (2024) into the related work and correct all noted typos (L224, L250, Figure 5c) in the final version.**

---

> > ### Comment · Reviewer_gigS · 2025-08-03
> >
> > Thank you for your detailed explanations, which have addressed most of my problems. I still need some clarifications about your rebuttal:
> >
> > 1. Can you provide more details about "2 questions per confidence bin across 5 bins from WebQA, NQ, and BioASQ (10 examples total), and asked 10 human raters whether they agreed with the user model’s judgment"? How do you define agreement between models and human raters?
> >
> > 2. It seems that increasing k does not improve ACC for your method but does improve it for the baseline. Could you explain this phenomenon? Is it possible that further increasing k can eliminate the gap?

---

> > > ### Author Response · Authors · 2025-08-04
> > >
> > > We sincerely thank the reviewer for actively engaging in the discussion and contributing valuable feedback to improve the quality of our work.
> > >
> > > ---
> > >
> > > # Answer to Question 1.
> > > Our human evaluation was conducted to quantitatively assess how well CalibRAG’s surrogate user model replicates actual human decision-making. Specifically, we sampled 10 questions (2 per confidence bin) from the WebQA, NQ, and BioASQ datasets.
> > >
> > > For each question, we presented the following four answer choices:
> > > 1. The answer generated by the surrogate user model
> > > 2. Three alternative answers generated from other documents (intentionally constructed to be potentially confusing)
> > >
> > > Human raters were provided with the question, the RAG context ($z$) generated from the document ($d$) selected by CalibRAG, the corresponding confidence score ($c$), and the four candidate answers. They were asked to choose the most convincing answer. We defined agreement as the proportion of cases where the human selected the same answer as the surrogate user model.
> > > A total of 10 human raters evaluated 10 questions each, resulting in 100 total judgments. Among them, 81.3% matched the surrogate user model's choice. This high level of agreement suggests that surrogate user model closely aligns with human judgment and that its evaluation pipeline meaningfully approximates real-world decision-making behavior.
> > >
> > > *If the paper is accepted, we will include the full human evaluation form and additional implementation details in the appendix to ensure transparency and reproducibility.*
> > >
> > >
> > > # Answer to Question 2.
> > > CalibRAG leverages confidence-aware reranking to prioritize the most informative documents at the top of the ranking. As a result, increasing the number of input documents $k$ provides only marginal performance gains. In fact, CalibRAG is already effective with just $k=1$, as it reliably identifies the most useful document for generation.
> > > In contrast, the baseline method (BM25 + Number) may initially rank less informative documents at the top. Therefore, increasing $k$ allows higher-quality documents to be included, which can improve performance. However, our experiments show that the performance of the baseline also plateaus around $k=5$ (see Table R.4).
> > > This phenomenon is well-explained by prior work. Liu et al. (2023) [1] demonstrate the “Lost in the Middle” effect, where LLMs tend to underutilize information from middle-position documents in long contexts, potentially degrading performance. Similarly, Kim et al. (2024) [2] report that adding more documents increases the risk of conflicting information and noise, which can hinder model accuracy.
> > > Moreover, feeding many documents simultaneously into an LLM is computationally expensive. CalibRAG avoids this inefficiency by evaluating each document independently and without requiring autoregressive decoding.
> > > In summary, CalibRAG achieves high accuracy and excellent calibration even with a small number of documents, offering clear advantages in both efficiency and practicality.
> > >
> > > **Table R.4.** Performance of the baseline (Number) at $k=5$.
> > >
> > > | Method         | AUROC |  ACC  |  ECE   |   BS   |
> > > |----------------|-------|-------|--------|--------|
> > > | Number (k=5)   | 65.07 | 41.19 | 0.2584 | 0.2856 |
> > >
> > > [1] Liu, Nelson F., et al. "Lost in the Middle: How Language Models Use Long Contexts." TACL., 2024.
> > >
> > > [2] Kim, Jaehyung, et al. "SuRe: Summarizing Retrievals using Answer Candidates for Open-domain QA of LLMs." ICLR., 2024.
> > >
> > > ---
> > >
> > > **We truly appreciate your thoughtful questions, which have helped us clarify key aspects of our study. We hope these exchanges contribute to a more complete and polished final version of our paper.**

---

> > > ### Author Response · Authors · 2025-08-07
> > >
> > > Dear Reviewer gigS,
> > >
> > > Thank you very much for your thoughtful engagement throughout the review process. Your comments and questions have been instrumental in helping us improve the clarity and quality of our work.
> > >
> > > We have done our best to address your follow-up questions in a sincere and detailed manner, and we hope that our responses have resolved your main concerns.
> > >
> > > As the author-reviewer discussion period is drawing to a close, we would greatly appreciate any final feedback you may have. If our responses have satisfactorily addressed your concerns, we would kindly and respectfully ask you to consider revisiting your initial rating.
> > >
> > > If there are any remaining questions or additional clarifications that would help you feel more confident in raising your score, please do not hesitate to let us know. We are fully committed to providing prompt and thorough responses, and to making any necessary improvements to meet your expectations.
> > >
> > > Your feedback is invaluable to the final quality of our paper, and we stand ready to make continued efforts until the very end.
> > >
> > > Once again, thank you for your time, careful consideration, and constructive feedback.
> > > We sincerely look forward to hearing from you.
> > >
> > > Sincerely,
> > >
> > > The Authors

---

> > > > ### Comment · Reviewer_gigS · 2025-08-08
> > > >
> > > > Thank you for the further explanation. The results of using more documents overall make sense to me. In the revised version, I would like to see a thorough human study demonstrating the agreement of your method with real users. I also suggest improving the clarity of the writing.
> > > >
> > > > As the rebuttal has addressed most of my concerns, I will raise my score.

---

> ### Author Response · Authors · 2025-08-08
>
> Dear Reviewer gigS,
>
> Thank you for your thoughtful feedback and for raising your score.
>
> We appreciate your suggestions and will reflect them in the revision.
>
> Sincerely,
>
> The Authors

---

### Official Review · Reviewer_FRsE · 2025-07-01

**Clarity:** 3
**Significance:** 3
**Originality:** 3
**Rating:** 5
**Confidence:** 4

**Summary:**

This paper studies to calibrate the decision-making process for retrieval augmented generation (RAG). The paper trains a forecasting function which takes a query-document pair and a temperature parameter representing user behavior. The model is trained on data generated by the LLM with different temperature setup. Experiments show that, not only the trained model can better calibrate the results, but it can also be used for context selection for RAG.

**Questions:**

- I wonder what the stop criterion is for the inference iterations? Or if step 3 will only be executed once, then what if there is still no quality retrieved document?
- Any intuitions on using Fourier positional encoding for temperature parameter? Are there other potential choices?

**Ethical Concerns:**

["NO or VERY MINOR ethics concerns only"]

**Final Justification:**

Overall, the paper studies a meaningful task and proposes a novel and solid approach. My concerns are addressed well during rebuttal and the explanation should be included in the next version.

**Limitations:**

yes

**Quality:**

4

**Strengths And Weaknesses:**

### Strengths
- The calibration problem for RAG is interesting and has its implications in real applications. It is interesting to see how confident/uncertain the LLM is based on the provided context and whether the response will be helpful.
- The proposed method is also sound and innovating. The usage of temperature to simulate user behavior and help data generation and training seems to be a promising approach.
- Some of the tricks and observations made by this work could motivate more research in similar directions.
- The paper is clearly written. I like the example in figure 2 which clearly demonstrates the task and motivation.

### Weaknesses
Overall I think it is a nice work and I didn't notice any overwhelming weakness.
- It would be beneficial to include some related works on RAG research in the background discussion.
- Some more discussions on limitations and future works could be beneficial for this paper as well as future researchers.

---

> ### Author Rebuttal · Authors · 2025-07-31
>
> >**[W1, W2] Can the paper provide a clearer positioning within existing RAG literature and discuss limitations and future directions more explicitly?**
>
> Thank you for the thoughtful suggestions. While we have included related RAG works in Appendix A.2, we recognize that their placement may reduce accessibility. We will revise the introduction to more clearly discuss RAG-related literature and better contextualize our contribution. Additionally, we will update the conclusion to include a concise discussion of limitations and potential future directions to support ongoing research in this area.
>
> >**[Q1] I wonder what the stop criterion is for the inference iterations? Or if step 3 will only be executed once, then what if there is still no quality retrieved document?**
>
> Thank you for the insightful question. To ensure a fair and consistent comparison across all methods, we did not perform Step 3 in any setting except for Figure 5-(c). In that experiment, we applied query reformulation only once, explicitly to evaluate its potential impact.
> We also note that query reformulation is an optional module in our pipeline. In real-world deployments, users can decide whether to enable it based on the trade-off between computational cost and performance gain.
> That said, we agree that if the reformulated query still results in low-quality retrieved documents, a single iteration may be insufficient. This limitation stems from both the inherent recall limitations of the retriever and the limited representational coverage of the document pool.
> To address this, incorporating adaptive multi-step retrieval mechanisms, possibly coupled with uncertainty-aware stopping criteria, is a promising direction for future work. We will clarify this detail in the final version of the paper.
>
> >**[Q2] Any intuitions on using Fourier positional encoding for the temperature parameter? Are there other potential choices?**
>
> Thank you for the question. Using a scalar $t$ directly often limits representational power and gradient flow, making it difficult for the model to condition on user behavior effectively. To address this, we adopt Fourier positional encoding, which maps $t$ into a high-dimensional space using sinusoidal functions. This approach is widely used in neural fields (e.g., NeRF [1]) to inject continuous inputs into deep models and enables smoother generalization across varying $t$. We found it effective and lightweight, though other encodings (e.g., learned embeddings) are also possible.
>
>
>
> [1] NeRF: Representing Scenes as Neural Radiance Fields for View Synthesis., Ben Mildenhall, et al., ECCV 2020.

---

> > ### Comment · Reviewer_FRsE · 2025-08-05
> >
> > Thanks for the response with explanation. I don't have other questions or concerns. I will keep my initial rating which aligns with my final judgement.

---

### Note · Authors · 2025-08-12

Dear Area Chair and Reviewers,

We sincerely appreciate the constructive feedback from all reviewers.

Our paper proposes the CalibRAG framework to address a previously overlooked problem in RAG systems: the alignment between model confidence and human decision accuracy.

Our core contribution is a user-aware forecasting function that predicts decision correctness by modeling user behavioral traits (e.g., risk tolerance) with a temperature parameter, $t$. This approach was recognized by multiple reviewers for its novelty and soundness.

We believe CalibRAG offers more than a simple performance improvement; it presents a new approach for building safer and more reliable human-AI decision-making systems.

Thank you once again for your valuable guidance.

Sincerely,

The Authors

---

### Decision · Program_Chairs · 2025-09-17

**Decision:**

Accept (poster)

**Comment:**

This paper introduces CalibRAG, a method for aligning confidence estimates in retrieval-augmented generation systems with correctness of responses. The key idea is to model user risk tolerance via sampling temperature, using an LLM to simulate decision-making under different tolerances. A forecasting model is then trained to predict calibrated confidence scores conditioned on the query, retrieved document, and sampling temperature, and is applied at inference to select documents that maximize calibration.

Reviewers found the approach sound and innovative (FRsE, gigS, xGBU), with clear exposition (urW9), comprehensive baselines (gigS, urW9), and a dataset that may benefit the broader community. Concerns raised during review included potential latency overhead (xGBU), questions about how well the proposed framework mirrors real user decision-making (xGBU, gigS), and possible bias in evaluation (gigS). However, reviewers agreed that these issues were addressed convincingly in the rebuttal. Incorporating rebuttal clarifications regarding latency, decision modeling, and evaluation fairness would strengthen the final version of the paper.